# Essential dynamic interdependence of FtsZ and SepF for Z-ring and septum formation in *Corynebacterium glutamicum*

Adrià Sogues [1], Mariano Martinez[1], Quentin Gaday[1], Mathilde Ben Assaya[1], Martin Graña [2], Alexis Voegele [3], Michael VanNieuwenhze [4], Patrick England[5], Ahmed Haouz [6], Alexandre Chenal [3], Sylvain Trépout [7], Rosario Duran [8], Anne Marie Wehenkel [1✉] & Pedro M. Alzari [1✉]

The mechanisms of Z-ring assembly and regulation in bacteria are poorly understood, particularly in non-model organisms. *Actinobacteria*, a large bacterial phylum that includes the pathogen *Mycobacterium tuberculosis*, lack the canonical FtsZ-membrane anchors and Z-ring regulators described for *E. coli*. Here we investigate the physiological function of *Corynebacterium glutamicum* SepF, the only cell division-associated protein from *Actinobacteria* known to interact with the conserved C-terminal tail of FtsZ. We show an essential interdependence of FtsZ and SepF for formation of a functional Z-ring in *C. glutamicum*. The crystal structure of the SepF–FtsZ complex reveals a hydrophobic FtsZ-binding pocket, which defines the SepF homodimer as the functional unit, and suggests a reversible oligomerization interface. FtsZ filaments and lipid membranes have opposing effects on SepF polymerization, indicating that SepF has multiple roles at the cell division site, involving FtsZ bundling, Z-ring tethering and membrane reshaping activities that are needed for proper Z-ring assembly and function.

[1] Unité de Microbiologie Structurale, Institut Pasteur, CNRS UMR 3528, Université de Paris, 75015 Paris, France. [2] Bioinformatics Unit, Institut Pasteur de Montevideo, Montevideo 11400, Uruguay. [3] Unité de Biochimie des Interactions Moléculaires, Institut Pasteur, CNRS, UMR 3528, 75015 Paris, France. [4] Department of Chemistry, Indiana University, Bloomington, IN 47405, USA. [5] Plate-forme de biophysique moléculaire, C2RT-Institut Pasteur, CNRS, UMR 3528, 75015 Paris, France. [6] Plate-forme de cristallographie, C2RT-Institut Pasteur, CNRS, UMR 3528, 75015 Paris, France. [7] Institut Curie, INSERM U1196, CNRS, UMR 9187, Université Paris-Sud, Université Paris-Saclay, 91405 Orsay, France. [8] Analytical Biochemistry and Proteomics Unit, Institut Pasteur de Montevideo & Instituto de Investigaciones Biológicas Clemente Estable, Montevideo, Uruguay. ✉email: anne-marie.wehenkel@pasteur.fr; pedro.alzari@pasteur.fr

The prokaryotic tubulin homolog FtsZ is at the heart of bacterial cytokinesis. At the cell division site, FtsZ protofilaments form a highly dynamic, membrane-bound structure, the Z-ring, which serves as a scaffold for the recruitment of the extra-cytoplasmic cell wall biosynthetic machinery. Despite its discovery >25 years ago[1], the exact molecular mechanisms of Z-ring assembly and regulation remain enigmatic[2]. In the best studied organisms, such as *Escherichia coli* and *Bacillus subtilis*, the action of several auxiliary proteins is necessary to positively or negatively regulate Z-ring formation (EzrA, ZapA-D[3–7]), to tether the structure to the membrane (FtsA, ZipA, SepF[8–11]), and to ensure proper subcellular (mid-cell) localization (SlmA, Noc, MinC/MinD[12–14]). Most of these proteins exert their functions by binding directly to the highly conserved FtsZ C-terminal domain (FtsZ$_{CTD}$)[15], which is separated from the core GTPase domain by an intrinsically disordered linker of variable length and sequence. Except for SepF, all the above positive and negative FtsZ regulators are missing in *Actinobacteria*[16], for which cell division mechanisms are largely unknown.

The *sepF* gene is found in the *dcw* (division and cell wall) cluster along with *ftsZ* and many essential genes for cell division[17]. In *B. subtilis*, SepF is a non-essential membrane-binding protein that co-localizes with FtsZ at mid-cell and is required for correct septal morphology as part of the late divisome[18,19]. In contrast to *B. subtilis*, *sepF* is an essential gene in *Mycobacterium smegmatis*[20] and in the cyanobacterium *Synechocystis*[21], both of which lack an identifiable homolog of *ftsA*. In *M. smegmatis* SepF localizes to the Z-ring in a FtsZ-dependent manner and has been shown to interact with the conserved C-terminal domain of FtsZ in yeast-two-hybrid assays[20]. Like FtsA, SepF has self-associating properties[22] and thus appears as a likely candidate for FtsZ membrane tethering in *Actinobacteria*. However, the observed assembly of SepF into stable 50 nm diameter ring polymers (alone or by bundling FtsZ protofilaments) seems to lack the dynamic oligomerization properties that are a recurrent feature of divisome and Z-ring interactors[23,24]. Indeed, increasing evidence suggests that membrane anchors are not just passive but active players of Z-ring dynamics and regulation. For instance, FtsA has a dual role both serving to tether FtsZ filament fragments to the membrane and exercising an antagonistic function on polymerization dynamics and directional assembly at mid-cell, a prerequisite to robust proto-ring assembly and subsequent inwards growth of new cell wall[10,24,25]. While in *B. subtilis* FtsA and SepF may have complementary and partially overlapping functions[11], we asked what happens in species where only SepF is present as a major Z-ring membrane anchor. Here, we provide mechanistic insights for the FtsZ-SepF interaction and its interdependency for Z-ring assembly and septum formation in *C. glutamicum*. We show that SepF has a complex dynamic role at the division site and that the ternary interaction between SepF, FtsZ, and the membrane, coupled to FtsZ polymerization dynamics, are all required for proper function and assembly.

## Results

**The essential role of SepF in *C. glutamicum*.** SepF from *M. smegmatis* was shown to be essential for viability and this protein was indeed proposed to be the unique membrane anchor for FtsZ in *Actinobacteria*[20]. However, an early study reported that the *sepF* gene was not essential in *Corynebacterium glutamicum*[26], which would argue against a main membrane tethering role for SepF. Several attempts at deleting *sepF* from *C. glutamicum* using either homologous recombination or gene disruption failed, suggesting that this gene might indeed be essential for bacterial survival. To deplete *sepF* we designed a conditional mutant strain ($P_{ino}$-*sepF*) where the transcription of *sepF* was uncoupled from

its physiological promoters by placing a transcriptional terminator just before the *sepF* gene, and by putting it under the control of the previously described *myo*-inositol repressible promoter ($P_{ino}$) of the inositol phosphate synthase Ino1 gene[27]. Down-stream effects were not expected as *sepF* is the last gene to be transcribed in the *dcw* cluster in *C. glutamicum*[28]. We observed a rapid depletion of SepF in the presence of 1% *myo*-inositol, while in its absence the SepF protein levels remained close to the level in the wild-type (WT) strain (Fig. 1a). The growth curves of the depleted $P_{ino}$-*sepF* and WT strains followed a similar pattern during the first 6 h, but after that point growth stopped for the depleted strain (Fig. 1b). When observed under the microscope a strong phenotype was seen from the first time point ($t = 3$ h), with elongated cells (Fig. 1c, d). At later time points branching was also seen, which corresponds to the formation of new poles at misplaced sites over the lateral walls of the bacterial cell and is a recurrent phenotype of mycobacterial cell division defects[29] or of *E. coli* cells with a misplaced peptidoglycan machinery[30,31]. At later points of the time course ($t = 12$ h or overnight cultures) cell lysis was frequently observed. This *sepF* depletion phenotype was rescued when the strain was complemented with a plasmid carrying an extra copy of *sepF* under the control of the $P_{tet}$ promoter (Supplementary Fig. 1), thus demonstrating the essentiality of *sepF* in *C. glutamicum*.

Using the fluorescent D-ala-D-ala analog (HADA[32]) to label newly incorporated peptidoglycan (PG), we showed that SepF depletion did not affect polar elongation (Fig. 1c and Supplementary Fig. 2). However, PG incorporation at mid-cell was lost and the cells were unable to form septa, thus showing that SepF is an essential component of the divisome in *Corynebacteria*. This absence of septa is clearly different from the SepF depletion phenotype in *B. subtilis*, where septa were present but largely deformed[19], suggesting that SepF homologs might have evolved different functions linked to the presence or absence of other auxiliary proteins such as FtsA. A phylogenetic analysis of bacterial SepF homologs shows that the proteins from *Firmicutes* and *Actinobacteria* do indeed fall into two clearly distinct groups (Supplementary Fig. 3) and suggests vertical inheritance with no horizontal transfer between both phyla. Interestingly, detectable FtsA homologs could not be identified in *Actinobacteria* nor in *Cyanobacteria* or some early-branching *Firmicutes*, which—together with the presence of SepF-like proteins (but not FtsA) in some archaeal lineages—would suggest an ancestral role for SepF in cell division.

Above we showed that septal PG synthesis was impaired in the absence of SepF, indicating that the cells could no longer assemble a functional divisome at mid-cell. As the Z-ring precedes septum formation, we asked what happened to FtsZ localization during depletion. We introduced mNeon-FtsZ as a dilute label under the control of $P_{gntK}$, a tight promoter that is repressed by sucrose and induced by gluconate[33] (Supplementary Fig. 4a, b). The minimal leakage of this promoter in sucrose was sufficient to give a signal for mNeon-FtsZ without affecting the phenotype, which remained wild-type-like (Supplementary Fig. 4c) and with a phenotype during SepF depletion comparable to that of the $P_{ino}$-*sepF* strain (Supplementary Fig. 5). mNeon-FtsZ was followed every 3 h during the depletion of SepF. As expected, mNeon-FtsZ localized to mid-cell at time point 0, when SepF was still present, in a typical "Z-ring" (Fig. 1e and Supplementary Fig. 5). From the following time point at 3 h, mNeon-FtsZ was completely delocalized into foci (possibly representing short filament structures) all over the cell, showing that SepF is indeed necessary for bringing FtsZ to the membrane to form a unique and functional Z-ring. Interestingly at 6 h the distribution of mNeon-FtsZ, although lost at mid-cell, appears to be clustered and not randomly distributed throughout the cell (Fig. 1f). This observation points to an as yet undiscovered FtsZ

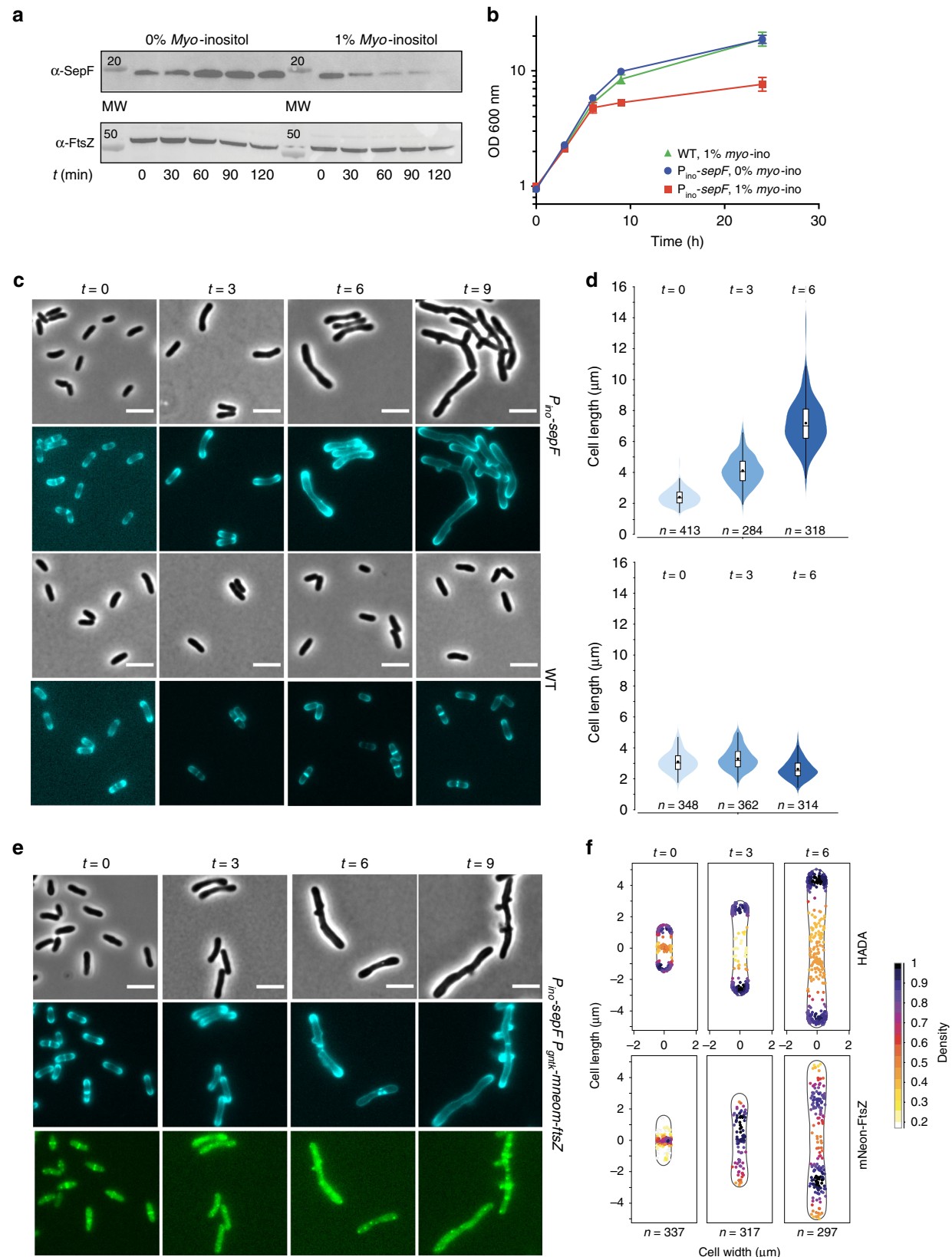

spatial regulation mechanism, since the well characterized nuclear occlusion and Min systems in *E. coli* and *B. subtilis* are absent in *Actinobacteria*[16]. Altogether these data show that in the absence of SepF, FtsZ cannot assemble at mid-cell anymore and is delocalized in the cell in a non-random manner.

**Molecular details of SepF interactions**. To understand the mechanisms by which SepF participates in early divisome assembly we set out to characterize the molecular details of the interaction of SepF with both the membrane and FtsZ. The molecular organization of SepF is highly conserved and, like FtsA

**Fig. 1 Phenotypic characterization of $P_{ino}$-sepF. a** SepF depletion. Western blots of whole-cell extracts from the $P_{ino}$-sepF strain, in the absence (not depleted) or presence (SepF depleted) of 1% myo-inositol during 2 h. SepF and FtsZ levels were revealed using anti-SepF (α-SepF) and anti-FtsZ (α-FtsZ) antibodies. Molecular weight (MW) markers are indicated. **b** Growth curves of WT cells in 1% myo-inositol (green triangles), $P_{ino}$-sepF in the absence (blue circles) or presence (red squares) of 1% myo-inositol. Error bars represent the mean ± SD. **c** Representative images in phase contrast (upper row) and HADA fluorescent signal (lower row) of $P_{ino}$-sepF and WT strains in 1% myo-inositol at time points 0, 3, 6, and 9 h after myo-inositol addition. Heatmaps representing the localization pattern of HADA at 0, 3, and 6 h are shown in Supplementary Fig. 2a; **d** Violin plot showing the distribution of cell length at time points 0, 3, 6 h after myo-inositol addition for $P_{ino}$-sepF (top) and WT (bottom) from **c**. The number of cells used in the analyses (n) is indicated below each violin representation, triplicate analyses and statistics are shown in Supplementary Fig. 2b and Supplementary Tables 5 and 6. The box indicates the 25th to the 75th percentile and the whiskers indicate the 95% confidence interval. The mean and the median are indicated with a dot and a line in the box, respectively. **e** Representative images in phase contrast (upper row), HADA fluorescent signal (middle row) and mNeon-FtsZ fluorescent signal (bottom row) at time points 0, 3, 6, and 9 h after myo-inositol addition for cells grown in minimal medium CGXII supplemented with 4% sucrose. **f** Heatmaps representing the localization pattern of HADA and mNeon-FtsZ at 0, 3, and 6 h. n numbers represent the number of cells used in the analyses. Triplicate analyses for the distribution of cell length at time points 0, 3, 6 h, as well as heatmaps for fluorescence distribution are shown in Supplementary Fig. 5. Scale Bars are 5 μm. Source data are provided as a Source Data file. The data shown are representative of experiments made independently in triplicate.

or ZipA, the protein contains an intrinsically disordered linker (L) of about 50 residues that separates the putative membrane-binding peptide (M) at the N-terminus from the FtsZ-binding core domain (C) at the C-terminus[11,20] (Fig. 2a). We proved that the predicted amphipathic helix at the N-terminus of *C. glutamicum* SepF did interact with lipid membranes (Supplementary Fig. 6a–c). Using tryptophan fluorescence titration, the peptide corresponding to the first 14 amino acids of SepF (SepF$_M$) was shown to bind small unilamellar vesicles (SUVs) with a Kd of 32 (+/−2) μM. In far-ultraviolet (UV) circular dichroism the SepF$_M$ peptide in solution behaved as a random coil and only folded into an α-helix upon interaction with SUVs, a behavior similar to that seen for *B. subtilis* SepF[11].

To elucidate the structural basis of FtsZ recognition, we crystallized the SepF C-terminal core domain (SepF$_{ΔML}$) in complex with a 10-residues peptide comprising the conserved C-terminal domain (CTD) of FtsZ (FtsZ$_{CTD}$, Fig. 2b) and determined the crystal structure at 1.6 Å resolution (Supplementary Table 1). The SepF structure revealed a symmetric homodimer, with monomers that contain a central four-stranded β-sheet stacked against two α-helices (α1, α2) involved in dimerization and capped by a C-terminal α-helix (α3) on the opposite side of the sheet (Fig. 2c). The homodimer contains two identical FtsZ$_{CTD}$-binding pockets, each made up of residues coming from both protomers (Fig. 2d), defining a dimeric functional unit for the SepF–FtsZ interaction. This 2:2 binding stoichiometry can explain mechanistically why *B. subtilis* SepF has a bundling effect on FtsZ protofilaments[34]. Corynebacterial SepF has a similar capability, as shown by FtsZ polymerization assays at different SepF concentrations (Fig. 2e). Even at sub-stoichiometric amounts of full-length SepF, the data showed an immediate influence on polymerization dynamics and a strong FtsZ bundling effect. Comparable changes on FtsZ polymerization were also observed for the C-terminal core alone (SepF$_{ΔML}$) but not for a SepF double mutant (SepF$_{K125E/F131A}$, see below) that is unable to bind FtsZ (Supplementary Fig. 7a, b), excluding the possibility that the light scattering signal could result from SepF polymerization alone. Furthermore, visualization of the protein mixture by negative stain electron microscopy (EM) after 10 min of incubation clearly showed thick bundles of FtsZ protofilaments, as well as highly curved filaments (Fig. 2f). Filaments and bundles were not observed in the absence of nucleotide or with SepF alone. Moreover, GTP and the slowly hydrolyzable analog GMPCPP produced a similar FtsZ bundling behavior, suggesting that GTP hydrolysis is not required for bundling (Supplementary Fig. 7c). We also showed that SepF did not significantly affect the GTPase activity of FtsZ (Supplementary Fig. 7d), in agreement with previous results on *B. subtilis* SepF[22].

The bound FtsZ$_{CTD}$ is clearly visible in the electron density map (Supplementary Fig. 8a), and adopts a hook-like extended conformation that fits into a mostly hydrophobic binding pocket formed by conserved SepF residues, where it is further stabilized by additional intermolecular hydrogen-bonding interactions (Fig. 2d and Supplementary Fig. 8b). The apparent Kd value for the SepF$_{ΔML}$–FtsZ$_{CTD}$ interaction, as determined by surface plasmon resonance (SPR), was 15 μM (+/−1 μM) (Supplementary Fig. 9), a value that is in the same range as those previously reported for other FtsZ$_{CTD}$ interactors such as ZipA[35], FtsA[36], or ZapD[37]. The interface was further validated by mutating two FtsZ$_{CTD}$-contact residues in SepF: F131A and K125E. Compared with the wild-type protein, the single mutant SepF$_{ΔML,F131A}$ was greatly compromised for FtsZ binding (apparent Kd = 340 +/− 47 μM), whereas the double mutant SepF$_{ΔML,K125E/F131A}$ exhibited no detectable binding in the range of protein-peptide concentrations tested (Fig. 2g and Supplementary Fig. 9).

Most FtsZ-binding proteins that have been characterized to date recognize the FtsZ$_{CTD}$, which represents a "landing pad" for FtsZ interactors[38]. Other known structures of regulatory proteins in complex with FtsZ$_{CTD}$ include *T. maritima* FtsA and the *E. coli* proteins ZipA, SlmA and ZapD[35–37,39]. These crystal structures had shown that the FtsZ$_{CTD}$ peptide can adopt multiple conformations depending on its binding partner, from full- or partial-helical states as in the FtsA or ZipA complexes to distinct extended conformations as in ZapD or SlmA. The SepF-bound structure of the FtsZ$_{CTD}$ peptide revealed yet another non-helical conformation, reflecting the large conformational space that this small, highly conserved sequence can adopt in different biological contexts. It is interesting to note that SlmA and SepF, despite their different structures and binding pockets, interact with the same highly conserved hydrophobic motif of the FtsZ$_{CTD}$ (Supplementary Fig. 10).

**Membrane and FtsZ binding in vivo.** To further evaluate the physiological roles of SepF-membrane and SepF–FtsZ interactions we constructed fluorescently tagged SepF constructs that were either unable to bind the lipid membrane (SepF$_{ΔML}$-Scarlet) or impaired for FtsZ binding (SepF$_{K125E/F131A}$-Scarlet). When SepF-Scarlet was overexpressed under the control of the $P_{gntK}$ promoter, it complemented the SepF depletion strain for growth (Fig. 3a, b and Supplementary Fig. 11). It should be noted, however, that the cells were more elongated than control cells, indicating that the C-terminal fusion does affect the function of SepF to some extent. In contrast, both SepF$_{ΔML}$-Scarlet and SepF$_{K125E/F131A}$-Scarlet completely failed to complement the strains and showed distinct localization patterns (Fig. 3c). Upon removal of the membrane-binding and linker domains, the SepF$_{ΔML}$-Scarlet construct was observed to form distinct foci of

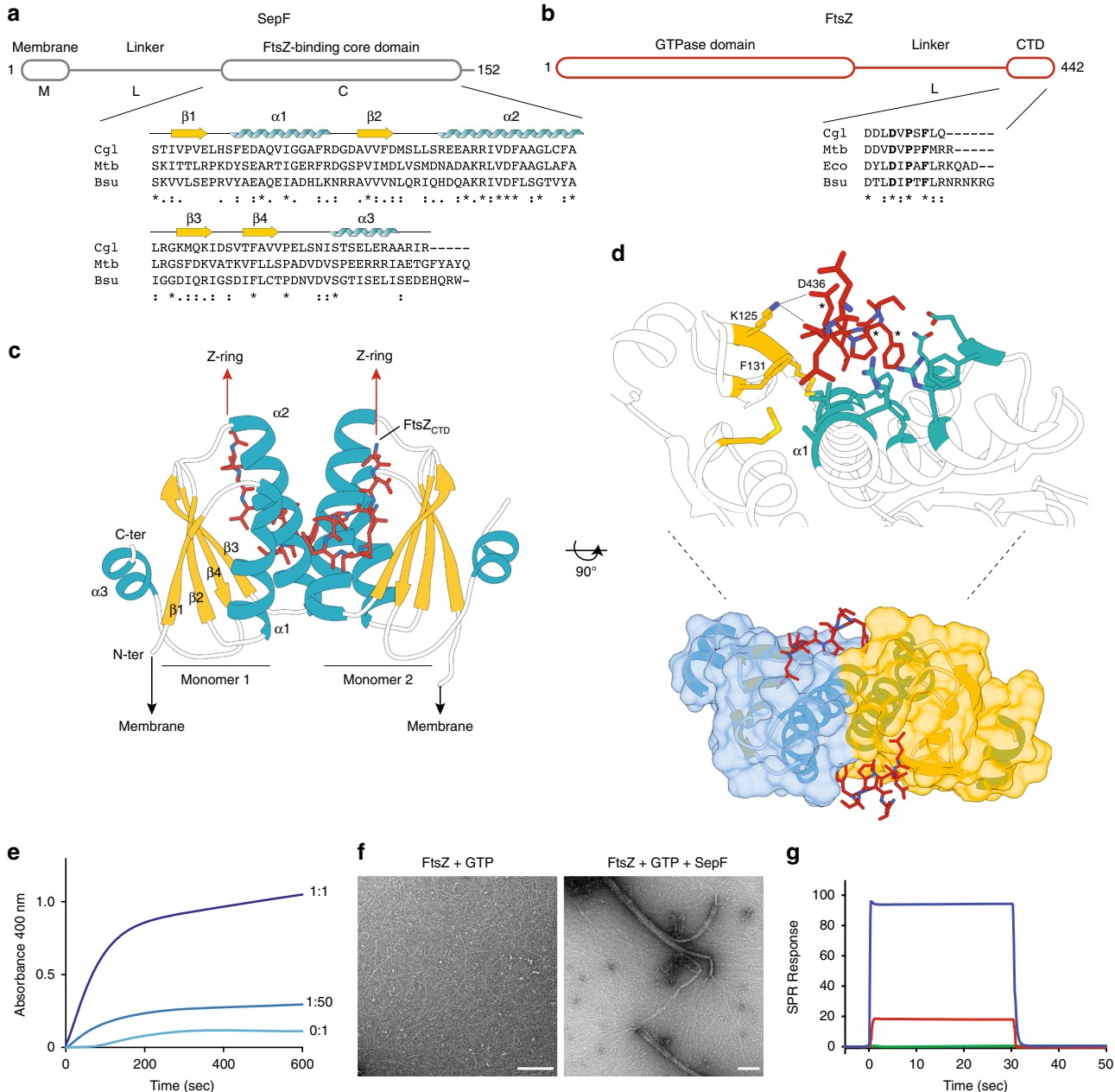

**Fig. 2 Molecular characterization of the SepF–FtsZ interaction. a** Schematic outline of SepF domains and sequence alignment of selected SepF homologs (Cgl, *C. glutamicum*; Mtb, *M. tuberculosis*; Bsu, *B. subtilis*). Secondary structure elements are shown above the sequences. **b** Schematic outline of FtsZ domains and FtsZ$_{CTD}$ sequence alignment of selected homologs (Eco, *E. coli*). Asterisks (*) indicate strictly conserved positions in the alignment and residues involved in SepF binding are shown in bold. **c** Crystal structure of the SepF dimer in complex with two FtsZ$_{CTD}$ peptides. The orientations of the N- and C- termini of SepF and FtsZ are compatible with membrane binding on one hand and Z-ring formation on the other. **d** Detailed view of the FtsZ-binding pocket in SepF showing residues involved in protein–protein interactions (see Supplementary Fig. 8 for details). Conserved FtsZ$_{CTD}$ residues D$_{436}$, P$_{438}$, and F$_{440}$ are labeled (*), and SepF residues K125 and F131 were those mutated to abolish FtsZ binding. **e** FtsZ polymerization in the presence of varying levels of SepF. The stoichiometric SepF:FtsZ ratios are indicated for each curve. **f** Negatively stained EM micrographs of FtsZ filaments in the absence (left) and presence (right) of SepF. The polymers in the left-hand panel have a width of about 4 nm, which corresponds to the width of FtsZ. The bundles shown in the right-hand panel range in width from 20–60 nm. Scale bars are 150 nm. The data shown are representative of experiments made independently in triplicate. **g** SPR responses in resonance units (RU) for 200 μM FtsZ$_{CTD}$ interacting with immobilized SepF (blue), SepF$_{F131A}$ (red), and SepF$_{K125E/F131A}$ (green). The detailed SPR results are shown in Supplementary Fig. 9. Source data are provided as a Source Data file.

SepF, similar to those seen for mNeon-FtsZ. This deletion mutant of SepF can still bind and bundle FtsZ as shown above but has lost the capacity to find the mid-cell and go to the membrane (Fig. 3c).

Importantly, the SepF$_{K125E/F131A}$-Scarlet construct containing the mutations that abolish FtsZ interaction was diffuse in the cytoplasm, showing that FtsZ binding is needed for membrane attachment even though the protein contains the N-terminal amphipatic helix. This suggests that, although the N-terminal peptide of SepF can bind membranes by itself (Supplementary Fig. 6), it would be unable to direct a protein monomer to the membrane. To confirm this hypothesis, we constructed a

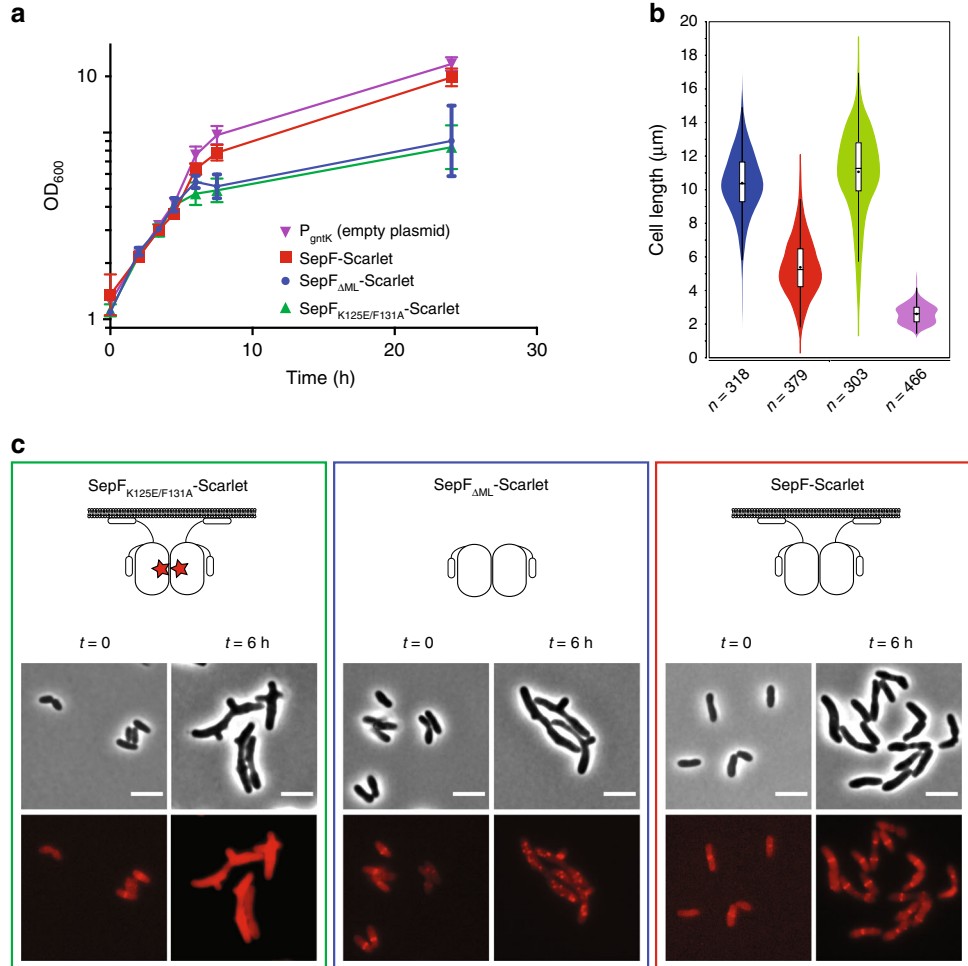

**Fig. 3 Complementation and localization of SepF mutants in the $P_{ino}$-sepF strain. a** Growth curves of SepF-Scarlet (red), SepF$_{\Delta ML}$-Scarlet (blue), SepF$_{K125E/F131A}$-Scarlet (green) expressed in the $P_{ino}$-sepF-$P_{gntK}$ background in 1% *myo*-inositol plus 1% gluconate and $P_{ino}$-sepF-$P_{gntK}$ strain in 0% *myo*-inositol plus 1% gluconate (purple). Error bars represent the mean ± SD. **b** Violin plot showing the distribution of cell length at time point 6 h after *myo*-inositol and gluconate addition for the strains corresponding to the growth curve (same color code). The number of cells used in the analyses (*n*) is indicated below each violin representation. The box indicates the 25th to the 75th percentile and the whiskers indicate the 95% confidence interval. The mean and the median are indicated with a dot and a line in the box, respectively. **c** Representative images in phase contrast (upper row) and Scarlet fluorescent signal (lower row) of the complemented strains of **a**. *t* = 0 corresponds to the strains before depletion by *myo*-inositol and induction of exogenous $P_{gntK}$ controlled constructs by gluconate. Western blots of whole-cell extracts from the above strains during depletion, as well as triplicate analyses for the distribution of cell length at time points 6 h are shown in Supplementary Fig. 11. The data shown are representative of experiments made independently in triplicate. Scale bars = 5 μm. Source data are provided as a Source Data file.

fusion protein in which the membrane-binding and linker regions of SepF (residues 1–63) were fused to the monomeric protein Scarlet. We found that this construct remained cytoplasmic when overexpressed in *C. glutamicum* (Supplementary Fig. 12), confirming that a higher avidity (such as that provided by FtsZ filaments decorated with multiple SepF dimers) may be needed for membrane tethering. Our results are consistent with previous work showing that SepF from *M. smegmatis* was dependent on FtsZ for localization at the Z-ring[20], indicating that a dynamic interplay between the two proteins is required for membrane association and Z-ring formation, and that SepF needs to act in the early actinobacterial divisome. This is also reminiscent of FtsA from *E. coli*, which requires FtsZ for localization at the septum[40], although FtsA can go to the membrane by itself[8]. Taken together, the above results demonstrate that SepF and FtsZ are intimately linked and interdependent to form a functional Z-ring and a viable cell in *Actinobacteria*.

**A putative mechanism for SepF polymerization**. The crystal structure of the globular core of SepF reported in this work differs from the available structures of other bacterial (pdb codes 3P04, 3ZIE) and archaeal (3ZIG, 3ZIH) SepF-like homologs in that it contains an additional helix (α3) at its C-terminus (Fig. 2c). This helix was predicted but not seen in other bacterial homologs because it is either missing in the construct or structurally disordered in the crystal[11]. Helix α3 formation and stabilization were not due to FtsZ binding, because the crystal structure of unliganded SepF$_{\Delta ML}$ (Supplementary Table 1) revealed the same overall structure than FtsZ$_{CTD}$-bound SepF$_{\Delta ML}$ (rmsd of 0.83 Å for 160 equivalent Cα atoms in the homodimer), indicating that peptide binding produced no significant conformational changes. The presence of this helix has important functional implications as it caps the β-sheet (Fig. 4a) that was previously described as the dimerization interface of *B. subtilis* SepF[11]. In fact, the crystal structure of the latter revealed two possible dimerization interfaces: one that occurred via the central β-sheet, which was

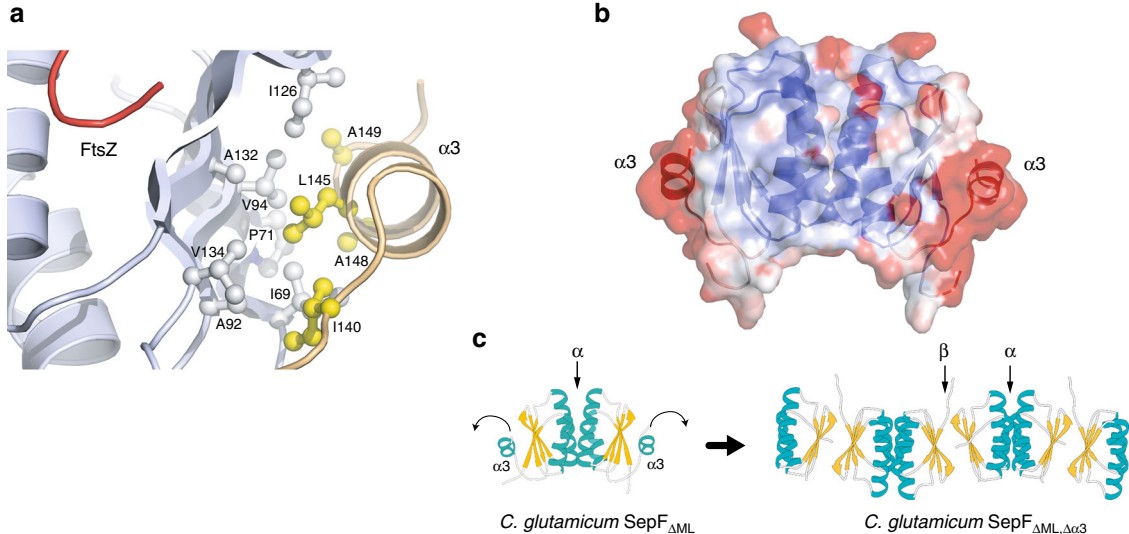

**Fig. 4 Putative regulatory role of helix α3. a** Hydrophobic interactions mediate the association of the C-terminal helix α3 (yellow) with the central β-sheet (gray) in the SepF protomer. **b** Overall structure of the unliganded SepF$_{\Delta ML}$ homodimer color coded according to temperature factors, from blue (lowest values) to red (highest values). **c** Deletion of helix α3 in SepF$_{\Delta ML,\Delta\alpha3}$ promotes the tight interaction between opposite β-sheets from different SepF dimers, leading to the formation of linear SepF polymers (see also Supplementary Fig. 13).

proposed to define the functional unit, and one that occurred via a 4-helix bundle, thought to be involved in protein oligomerization and ring formation[11] (Supplementary Fig. 13a). Our crystal structure superimposed well with the 4-helix bundle-mediated SepF dimers from *B. subtilis*, supporting the hypothesis that the *C. glutamicum* SepF homodimer (Fig. 2c) is conserved across different bacterial species.

If the functional SepF unit is conserved, the β-sheet-mediated interface observed in *B. subtilis* SepF crystals could therefore mediate protein polymerization in solution. This interface, which was also seen in SepF-like proteins from *Pyrococcus furiosus* and *Archaeoglobus fulgidus*[11], is precluded in the *C. glutamicum* SepF structure by the presence of the C-terminal helix α3 (Fig. 4a). However, inspection of the unliganded SepF$_{\Delta ML}$ structure revealed that this helix displays considerably higher B-factor values than the rest of the protein (Fig. 4b) and a similar trend is also observed in the SepF$_{\Delta ML}$–FtsZ$_{CTD}$ complex, suggesting that helix α3 could play a regulatory role on SepF polymerization by uncovering the outer face of the β-sheet for intermolecular interactions. To further investigate this hypothesis, we removed the helix and crystallized the resulting SepF$_{\Delta ML,\Delta\alpha3}$ construct alone and in complex with FtsZ$_{CTD}$ (Supplementary Table 1). Despite a different crystal packing, the two structures did show a dimer-dimer association mediated by the opposing β-sheets in the crystal, generating linear SepF polymers similar to those observed for the *B. subtilis* homolog (Fig. 4c and Supplementary Fig. 13). These observations demonstrate that the β-sheet is indeed prone to self-interaction in solution and suggest that the amphipathic helix α3 might regulate SepF polymerization. It is interesting to note, however, that the anti-parallel orientation of the interacting β-sheets in three crystal forms of *C. glutamicum* SepF (Supplementary Fig. 13), although identical to each other, differs from the parallel orientation seen in the *B. subtilis* SepF structure, thus pointing to possible species-specific polymerization mechanisms.

The above polymerization model can account for the formation of SepF rings in *B. subtilis*[22]. In our hands, *C. glutamicum* SepF devoid of helix α3 (i.e., with the β-sheet accessible for protein–protein interactions) remained dimeric in solution and we were unable to detect higher order assemblies by

size exclusion chromatography or analytical ultracentrifugation (Supplementary Fig. 14). Furthermore, we could observe regular rings of ~ 40 nm diameter in negative stain EM for the related SepF homolog from *M. tuberculosis* (Supplementary Fig. 15), but our extensive attempts at detecting similar ring-like structures of *C. glutamicum* SepF were unsuccessful. For the latter, oligomerization may therefore require an increased local concentration, as found for instance in crystallization conditions or in a physiological situation linked to lipid membrane and/or FtsZ filament interactions, where avidity effects can push the equilibrium towards an oligomeric state. Taken together, the above biophysical data suggest that the C-terminal helix α3 could regulate SepF polymerization by controlling β-sheet-mediated interactions during Z-ring formation and divisome assembly.

**Reversible SepF oligomerization.** We next investigated the effects of lipid membranes and FtsZ binding on SepF polymerization. We found that incubation of SepF with SUVs led to SepF concentration-dependent polymerization in turbidity assays (Fig. 5a and Supplementary Fig. 16) and to the formation of large structures in dynamic light scattering (DLS) experiments (Supplementary Fig. 17a–c). When we looked at the end-point of these reactions in negative stain EM the lipid vesicles were tubulated (Fig. 5a), indicating that the N-terminal amphipathic helix of SepF has the capability to induce membrane remodeling upon protein polymerization. Surprisingly this behavior was completely reversed when the FtsZ$_{CTD}$ peptide was added to the reaction (Fig. 5b and Supplementary Fig. 17d): SepF depolymerized, lipid vesicles recovered their initial size in DLS, and small regular vesicles with no tubulation were observed in electron micrographs. When the same experiment was carried out with SepF$_{\Delta\alpha3}$, lacking the regulatory C-terminal helix, polymerization and vesicle tubulation were also seen but could only be partially reversed after addition of the FtsZ$_{CTD}$ peptide (Supplementary Fig. 18). In summary, SepF is a dimeric protein prone to polymerize forming rings at high concentration. Membrane binding would then displace the equilibrium to favor SepF polymerization in a concentration-dependent manner, while FtsZ binding would do it in the opposite sense, towards depolymerization.

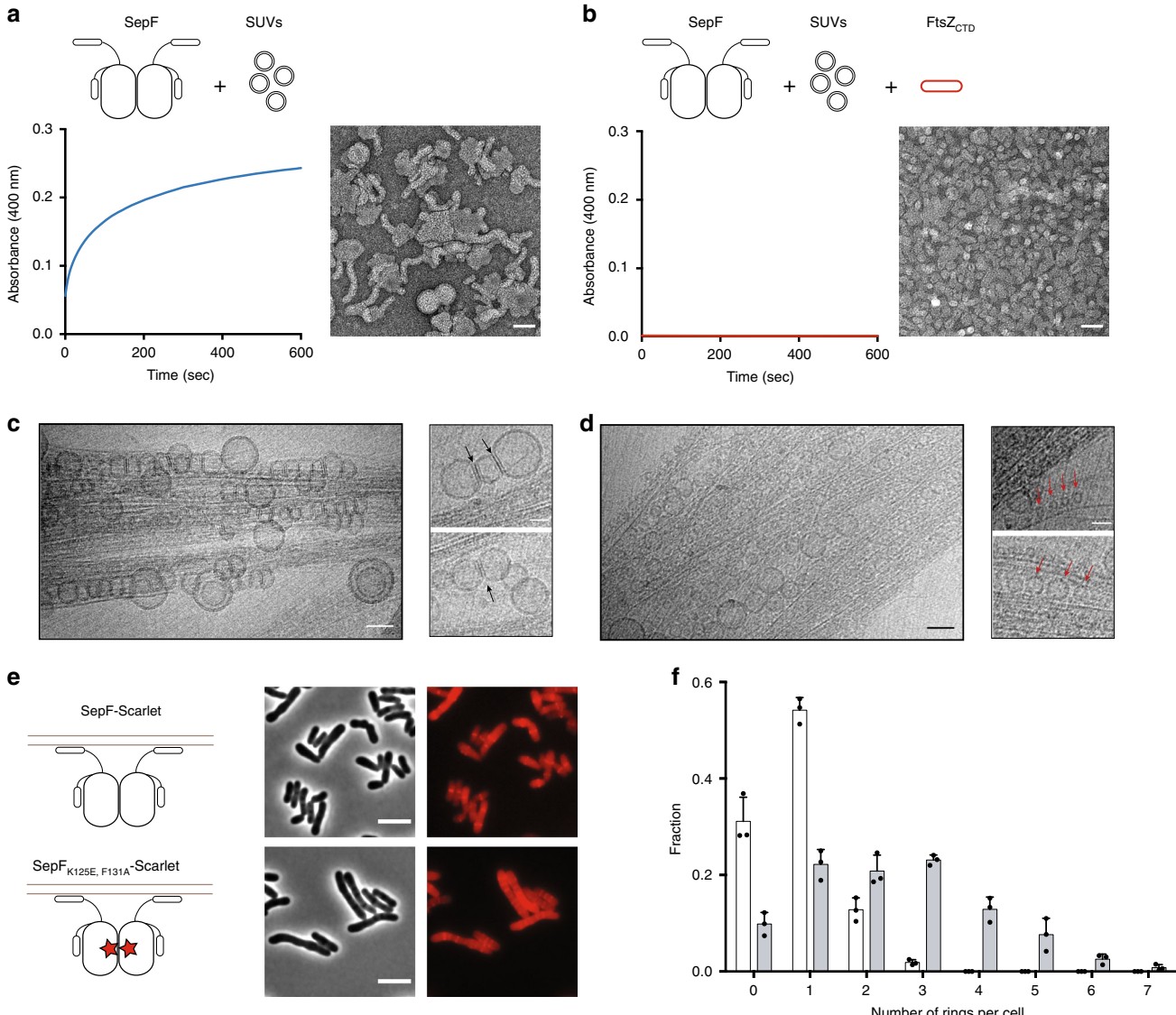

**Fig. 5 SepF oligomerization upon lipid membrane-binding and reversal by FtsZ. a** Polymerization assay and negatively stained EM images of the end-point of the assay for SepF in the presence of SUVs. Scale bar: 50 nm. **b** Idem for SepF in the presence of SUVs + FtsZ$_{CTD}$. Scale bar: 50 nm. **c** Cryo-EM images of the ternary complex full-length FtsZ-GTP + SepF + SUVs. A bundle of FtsZ filaments is decorated by tubulated lipid vesicles, mostly found on the periphery of the bundles. The smaller panels show two examples of tubulated lipid vesicles, where black arrows indicate the putative SepF rings. Scale bars in large and small panels, respectively: 50 and 30 nm. **d** In the central regions of the bundles, several small, non-tubulated, vesicles are in close contact with FtsZ filaments. Red arrows indicate a few examples of these vesicles in the right panels. Scale bars in large and small panels, respectively: 50 and 30 nm. **e** Phase contrast (left columns) and Scarlet fluorescent signal (right column) images show the phenotypic differences observed upon overexpression of SepF-Scarlet (top) and SepF$_{K125E/F131A}$-Scarlet (bottom) in the WT strain, where endogenous SepF is present. Western blots of whole-cell extracts from the above strains as well as triplicate analyses for the distribution of cell length are shown in Supplementary Fig. 19. Scale bars = 5 μm. **f** Frequency histogram indicating the number of ring-like structures per cell for SepF-Scarlet (white) and SepF$_{K125A/F131A}$-Scarlet (gray). The frequencies were calculated from *n* cells imaged from three independent experiments for each strain (For SepF-Scarlet, *n* = 423, 471, and 477; and for SepF$_{K125A/F131A}$-Scarlet, *n* = 336, 353, and 392). The bars, with data points overlapping, represent the mean ± SD. Source data are provided as a Source Data file. All data shown are representative of experiments made independently in triplicate.

In cryo-EM experiments with full-length FtsZ (instead of FtsZ$_{CTD}$), we were able to trap an intermediate state of the ternary complex (lipid membranes, SepF, FtsZ-GTP). In the images (Fig. 5c) negatively charged lipid vesicles of different sizes are brought together by rings of SepF, forming tubular structures at the periphery of the bundles, in which the SepF rings involved do not interact with FtsZ filaments. On the other hand, both these peripheral tubular structures (Fig. 5c) as well as the individual smaller vesicles trapped at the central bundle regions (Fig. 5d) attach to and decorate the FtsZ filaments, demonstrating that the

dimeric form of SepF retains lipid-binding capability. Taken together, these results point to a dynamic equilibrium between two SepF subpopulations, a dimeric form associated with FtsZ bundling and membrane tethering and a second polymeric form associated with membrane remodeling.

Further evidence for the interdependence of SepF and FtsZ dynamics in vivo was obtained by overexpressing SepF$_{K125E/F131A}$-Scarlet (which is unable to bind FtsZ) in the WT background. From mass spectrometry data (immunoprecipitation experiments coupled to crosslinking) we found that FtsZ

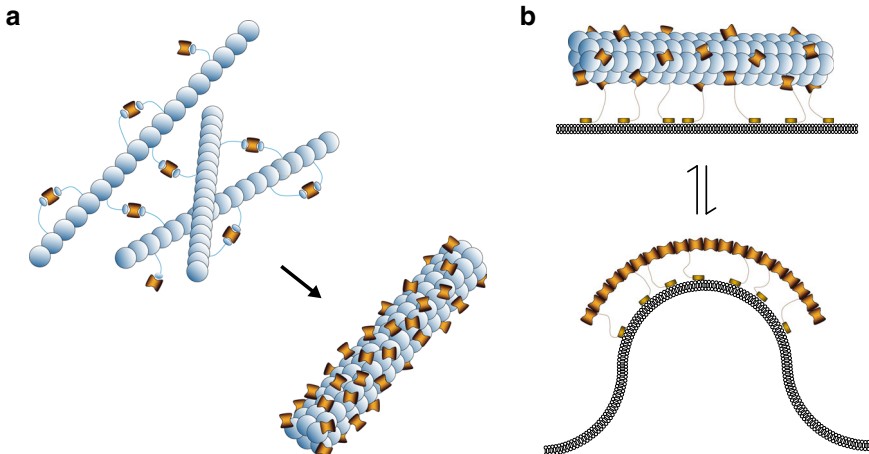

**Fig. 6 Schematic model depicting the possible mode of action of SepF in the early stages of Z-ring assembly and septum formation. a** Formation of the FtsZ-SepF complex in the cytoplasm leads to FtsZ filament bundling. **b** At the membrane, homodimeric SepF molecules bound to FtsZ filaments would sustain membrane tethering (top), while membrane-bound polymerized SepF would play an active role in remodeling the lipid bilayer for septum formation (bottom).

binding in the strain expressing SepF$_{K125E/F131A}$-Scarlet was greatly reduced (7.2-fold; see Methods section and Supplementary Table 2) compared to SepF-Scarlet, but not abolished, showing that the mutant and endogenous SepF proteins interacted with each other. When we looked at the localization of SepF in these two strains, we observed a net increase in the number of ring-like structures per cell for SepF$_{K125E/F131A}$-Scarlet as opposed to mostly single rings at mid-cell for SepF-Scarlet (Fig. 5e, f and Supplementary Fig. 19). Thus, a corrupted FtsZ-binding site led to the formation of multiple, more stable SepF rings that contain not only fully functional endogenous SepF but also non-functional SepF$_{K125E/F131A}$-Scarlet. Multiple-rings are not due to protein overexpression or to the presence of a fluorescent tag, because the multiple-rings phenotype is not observed for cells overexpressing wild-type SepF under the control of the strong $P_{tac}$ promoter (~ 30-fold protein overexpression upon induction), nor for cells in which the Scarlet tag was fused to the genomic copy of *sepF* under the control of its endogenous promoter (Supplementary Fig. 20). Since recombinant SepF$_{K125E/F131A}$ levels exceed the endogenous protein levels (Supplementary Fig. 19), these partially functional rings are expected to contain patches that cannot bind and stabilize FtsZ filaments. This would in turn interfere with the formation of a fully dynamic oligomeric Z-ring structure, which requires correct alignment and stabilization for solid treadmilling-driven assembly of the division machinery[10,41,42].

## Discussion
Here, we demonstrated that SepF and FtsZ are interdependent to form a functional Z-ring and that SepF is essential in *C. glutamicum*. Our structural data revealed the FtsZ-binding pocket, defining the SepF homodimer as the functional unit, and suggesting a reversible β-sheet-mediated oligomerization interface possibly regulated via an alpha helical switch. We found that FtsZ filaments and lipid membranes have opposing effects on SepF polymerization, pointing to a complex dynamic role of the protein at the division site, involving FtsZ bundling, Z-ring tethering and membrane reshaping activities that are needed for proper Z-ring assembly and function.

Our results put forward a mechanistic model for SepF function during the early stages of divisome assembly (Fig. 6). At cellular concentrations in the nanomolar range, unbound SepF dimers would neither bind to the membrane nor polymerize. A higher local SepF concentration would be achieved by FtsZ-GTP filament binding (Fig. 6a), which would in turn promote filament

bundling (because of the 2:2 stoichiometry of the FtsZ:SepF complex), while precluding SepF polymerization. This initial process would result in cytoplasmic FtsZ filaments and bundles presumably decorated with most available SepF molecules in the cell, as the molar concentration of SepF in the cell is considerably lower (5–20-fold) than that of FtsZ (Supplementary Fig. 21). At the membrane, oligomeric and non-oligomeric (i.e., dimeric) forms of SepF would coexist, as suggested by the cryo-EM images of the ternary complex (Fig. 5c, d). For instance, some SepF molecules would remain in a dimeric form attached to the bundles, providing enough amphipathic peptides to sustain membrane tethering (Fig. 6b, top), while others would detach from the bundles to form membrane-bound polymers that will play an active role in remodeling the lipid bilayer for septum formation (Fig. 6b, bottom). Such redistribution is made possible by the highly dynamic nature of the FtsZ–SepF-membrane interactions and the opposite effects of competitive filament/membrane binding on SepF oligomerization. How the SepF–FtsZ complexes are specifically localized to the mid-cell and whether there are additional regulatory factors affecting the oligomerization state of SepF remains to be elucidated. These could be unidentified divisome components, or else post-translational modifications such as protein phosphorylation, known to play an important role in the regulation of actinobacterial cell division and morphogenesis[43].

Protein dynamics are intrinsic functional features of tubulin (dynamic instability[44]) and FtsZ (treadmilling[10]). FtsZ by itself has the properties not only to self-organize and provide directionality, but also to deform lipids when a membrane-binding motif is attached to the protein[45–47]. In this case, FtsZ has been shown to assemble into dynamic vortices in vitro without the need for accessory proteins, but critically relying on concentration thresholds[46]. Concentration dependence is also true for accessory proteins such as the bundling protein ZipA[48], or ZapA[49] where the effect on FtsZ dynamics heavily relies on the protein concentrations used. In the bacterial cell FtsZ does not contain a membrane-binding domain and the self-assembly is counteracted by several parameters such as molecular crowding[50,51], as well as spatial and temporary constraints for mid-cell localization. A functional tethering system thus needs to allow for accumulation of enough FtsZ molecules (i.e., via bundling) without trapping these bundles in static states. In *E. coli*, it was recently shown that the bundling protein ZapA increases FtsZ filament stability without affecting the treadmilling activity[51]. The Z-ring also

needs to constantly adapt to the dynamics and shrinking of the membrane invagination during septal closure, and dynamic exchange is thus primordial to all FtsZ interacting systems.

Membrane remodeling activity has also been described for the *E. coli* membrane tether FtsA[25]. As FtsA and all other known FtsZ$_{CTD}$ interactors are absent from corynebacterial genomes, it is possible that SepF could play a more complex role in those organisms and contribute to both membrane tethering and remodeling. For SepF a plausible scenario could be that, during the early stages of assembly, polymerized FtsZ fragments are bundled and tethered to the membrane mainly by dimeric SepF, which—possibly assisted by auxiliary regulatory factors yet to be identified—would help FtsZ polymers to stabilize, find directionality, and start treadmilling to form a functional Z-ring. At the same time, this process would increase the local concentration of SepF at the membrane. Treadmilling would remove available FtsZ$_{CTD}$-binding sites, leading to SepF polymerization and membrane invagination, contributing to the net force required for cell constriction[52]. A possible consequence of this model is that SepF-induced septum formation would only occur when enough FtsZ has been accumulated at the membrane and treadmilling starts, making of SepF a checkpoint protein that would initiate constriction only once the cytomotive machinery is fully functional.

## Methods

**Bacterial strains and growth conditions.** All bacterial strains used in this study are listed in the Supplementary Table 3. *Escherichia coli* DH5α or CopyCutter EPI400 were used for cloning and were grown in Luria-Bertani (LB) broth or agar plates at 37 °C supplemented with 50 µg/ml kanamycin when required. For protein production, *E. coli* BL21 (DE3) was grown in LB or 2YT broth supplemented with 50 µg/ml kanamycin at the appropriate temperature for protein expression.

*Corynebacterium glutamicum* ATCC13032 was used as a wild-type (WT) strain. *C. glutamicum* was grown in brain heart infusion (BHI) at 30 °C and 120 rpm and was supplemented with 25 µg/ml kanamycin when required. When specified, minimal medium CGXII[53] supplemented with sucrose, gluconate, IPTG, tetracycline, kanamycine and/or *myo*-inositol was used.

**Cloning for recombinant protein production in *E. coli*.** The genes encoding for *C. glutamicum sepF* (*cg2363*), *M. tuberculosis sepF* (*Rv2147c*), and *C. glutamicum ftsZ* (*cg2366*) were codon optimized and synthesized for *E. coli* protein production (Genscript) and used as templates for subsequent cloning. They were cloned into a pT7 vector containing an N-terminal 6xHis-SUMO tag. SepF mutants containing different domains: SepF$_{ΔML}$ (amino acids 63 to 152), SepF$_{ΔML,Δα3}$ (amino acids 63 to 137), SepF$_{Δα3}$ (amino acids 1 to 137), SepF$_{ΔML,F131A}$, SepF$_{ΔML,K125E/F131A}$, SepF$_{K125E/F131A}$ and *M. tuberculosis* SepF$_{ΔML}$ (*Mtb*SepF$_{ΔML}$, comprising amino acids 122 to 218) were constructed using either pairs of complementary primers carrying the desired deletion or point mutations on the primers listed in the Supplementary Table 4. PCR products were digested with DpnI and transformed into chimio-competent *E. coli* cells. All plasmids were verified by Sanger sequencing (Eurofins Genomics, France).

**Cloning for recombinant protein expression in *C. glutamicum*.** For ectopic recombinant expression of the different constructs in *C. glutamicum*, we used the synthetic pTGR5 shuttle expression vector under the control of the $P_{tac}$ promoter, as well as two other expression vectors, pUMS_3 and pUMS_40, in which the $P_{tac}$ promoter of pTGR5 was, respectively, exchanged by the promoters $P_{gntK}$ of *C. glutamicum* or $P_{tet}$ from pCLTON1 vector[53–55] (Supplementary Table 3). FtsZ and SepF variants were assembled in these plasmids by either Gibson assembly or site-directed mutagenesis using the primers listed in Supplementary Table 4. For cellular localization studies, mScarlet-I[56] was fused on the C-terminal side of SepF and mNeonGreen[57] was fused on the N-terminal side of FtsZ, including in both cases a flexible linker between the two fused proteins.

For co-expression of SepF and FtsZ, plasmid pUMS3-mNeon-cgFtsZ was digested with *XbaI* and *SpeI* and the fragment containing PgntK-mNeon-cgFtsZ was ligated in pTGR5-cgSepF plasmid previously digested with *SpeI*.

**sepF conditional mutant and the sepF:scarlet strain.** To obtain the conditional depletion of *sepF*, the endogenous *sepF* was placed under the control of a repressible promoter. Using the two-step recombination strategy with the pk19mobsacB plasmid, we inserted the native promoter of the inositol phosphate synthase gene (*ino1– cg3323*), which can be repressed in the presence of *myo*-inositol[27]. A terminator followed by the ino1 promoter was amplified by PCR from

pk19-P3323-*lcpA*[58]. The 500 bp up-stream region and down-stream region of *sepf* were amplified using chromosomal DNA of *C. glutamicum* ATCC13032 as a template. The different fragments were assembled in a pk19mobsacB backbone by Gibson assembly (NEB). The plasmid was sequenced and electroporated into WT *C. glutamicum* ATCC13032. Positive colonies were grown in BHI media supplemented with 25 µg/ml kanamycin overnight at 30 °C and 120 rpm shaking. The second round of recombination was selected by growth in minimal medium CGXII plates containing 10% (w/v) sucrose. The insertion of the *ino1* promoter was confirmed by colony PCR and sequenced (Eurofins, France). All the oligonucleotides used in order to obtain and check this strain are listed in the Supplementary Table 4.

For the construction of the *sepF:scarlet* strain, we fused at the 3′ of the endogenous *sepF* the gene encoding for *scarlet* spaced by a linker (LEGSGQGPGSGQGSGH). We used the pk19mobsacB strategy, in which *sepF* coding region with scarlet and 500 bp down-stream of *sepF* were amplified from the pUMS4sepF-scarlet and chromosomal DNA of C. glutamicum ATCC13032 as a template, respectively. PCR fragments were assembled into the pk19mobsacB backbone by Gibson assembly obtaining the plasmid pk19-*sepF:scarlet*. The plasmid was sequenced and electroporated into WT *C. glutamicum* ATCC13032. Positive colonies were grown in BHI media supplemented with 25 µg/ml kanamycin overnight at 30 °C and 120 rpm shaking. The second round of recombination was selected by growth in BHI plates containing 10% (w/v) sucrose. The insertion of *scarlet* was confirmed by colony PCR and sequenced (Eurofins, France).

**Growth curves.** All strains were plated in CGXII media plates with 4% (w/v) sucrose as a carbon source for 2 days at 30 °C and then single colonies were inoculated in 10 ml of CGXII media 4% (w/v) sucrose overnight at 30 °C and 120 rpm shaking. The next day, 20 ml of CGXII media supplemented with the appropriate repressor or inducer were inoculated with the overnight cultures to a starting OD$_{600}$ of 1. OD$_{600}$ measurements were taken every 1.5 h. Each growth curve represents the average of three different growth curves originally from three different single colonies. For each time point a sample for western blot was taken. When required, CGXII media 4% (w/v) sucrose was supplemented with either 1% (w/v) *myo*-inositol, 50 ng/ml tetracycline or 1% (w/v) gluconate and 25 µg/ml kanamycin.

**Protein expression and purification.** N-terminal 6xHis-SUMO-tagged SepF, and derivate mutants (both from *C. glutamicum* and *M. tuberculosis*) were expressed in *E. coli* BL21 (DE3) following an autoinduction protocol[59]. After 4 h at 37 °C cells were grown for 20 h at 20 °C in 2YT complemented autoinduction medium containing 50 µg/ml kanamycin. Cells were harvested and flash frozen in liquid nitrogen. Cell pellets were resuspended in 50 ml lysis buffer (50 mM Hepes pH 8, 300 mM NaCl, 5% glycerol, 1 mM MgCl$_2$, benzonase, lysozyme, 0.25 mM TCEP, EDTA-free protease inhibitor cocktails (ROCHE) at 4 °C and lysed by sonication. The lysate was centrifuged for 30 min at 30,000 × *g* at 4 °C. The cleared lysate was loaded onto a Ni-NTA affinity chromatography column (HisTrap FF crude, GE Healthcare). His-tagged proteins were eluted with a linear gradient of buffer B (50 mM Hepes pH 8, 300 mM NaCl, 5% glycerol, 1 M imidazole). The eluted fractions containing the protein of interest were pooled and either dialyzed directly (for SPR experiments) or dialyzed in the presence of the SUMO protease (ratio used, 1:100). Dialysis was carried out at 4 °C overnight in 50 mM Hepes pH 8, 150 mM NaCl, 5% glycerol, 0.25 mM TCEP. Cleaved His-tags and His-tagged SUMO protease were removed with Ni-NTA agarose resin. The cleaved protein was concentrated and loaded onto a Superdex 75 16/60 size exclusion (SEC) column (GE Healthcare) pre-equilibrated at 4 °C in 50 mM Hepes pH 8, 150 mM NaCl, 5% glycerol. The peak corresponding to the protein was concentrated, flash frozen in small aliquots in liquid nitrogen and stored at −80 °C.

Codon optimized N-terminal 6xHis-SUMO-tagged *C. glutamicum* FtsZ was produced and purified as described above, except that KCl was used instead of NaCl and a TALON FF crude column (GE Healthcare) was used for affinity chromatography. All purified proteins used in this work have been run on an sodium dodecyl sulfate–polyacrylamide gel electrophoresis (SDS-PAGE) and are represented in Supplementary Fig. 22.

**Crystallization.** Crystallization screens were performed for the different SepF constructs and SepF–FtsZ$_{CTD}$ complexes using the sitting-drop vapor diffusion method and a Mosquito nanolitre-dispensing crystallization robot at 18 °C (TTP Labtech, Melbourn, UK). Optimal crystals of SepF$_{ΔML}$ (13 mg/ml) were obtained after one week in 10% PEG 8000, 0.2 M Zinc acetate, 0.1 M sodium acetate. The complex SepF$_{ΔML}$–FtsZ$_{CTD}$ (DDLDVPSFLQ, purchased from Genosphere) was crystallized at 10 mg/ml SepF$_{ΔML}$ and 5.8 mg/ml of FtsZ$_{CTD}$ (1:5 molar ratio) after 2 weeks in 100 mM sodium acetate pH 4.6, and 30% w/v PEG 4000. SepF$_{ΔML,Δα3}$ (17 mg/ml) crystallized within 2 weeks in 0.1 M MES pH 6, 20%w/v PEG MME 2000 and 0.2 M NaCl. The SepF$_{ΔML,Δα3}$–FtsZ$_{CTD}$ complex was crystallized at 17 mg/ml SepF$_{ΔML,Δα3}$ and 9.8 mg/ml FtsZ$_{CTD}$ (1:5 molar ratio) within 2 weeks in 0.1 M MgCl$_2$, 0.1 M MES pH 6.5 and 30% w/v PEG 400 buffer. Crystals were cryo-protected in mother liquor containing 33% (vol/vol) ethylene glycol or 33% (vol/vol) glycerol.

**Data collection, structure determination, and refinement.** X-ray diffraction data were collected at 100 K using beamlines ID30B and ID23-1 (wavelength = 0.97625 Å) at the ESRF (Grenoble, France). All datasets were processed using XDS[60] and AIMLESS from the CCP4 suite[61]. The crystal structures were determined by molecular replacement methods using Phaser[62] and *C. glutamicum* SepF (PDB code 3p04) as the probe model. In the case of SepF$_{\Delta ML}$, the structure was also independently determined by single-wavelength anomalous diffraction (SAD) phasing using Patterson methods to localize the protein-bound Zn ions (present in the crystallization buffer) with SHELXD[63] and automatic model building with Buccaneer from the CCP4 suite[64]. All structures were refined through iterative cycles of manual model building with COOT[65] and reciprocal space refinement with BUSTER[66]. Highly anisotropic diffraction was observed for SepF$_{\Delta ML,\Delta\alpha3}$ crystals, in which one of the two monomers in the asymmetric unit was largely exposed to solvent (see Supplementary Fig. 13c), and exhibited unusually high-temperature factors (average B values for all main-chain atoms of the two monomers were, respectively, 17 Å$^2$ and 58 Å$^2$). The crystallographic statistics are shown in Supplementary Table 1 and representative views of the final electron density map for each structure are shown in Supplementary Fig. 23. Structural figures were generated with Chimera[67] or Pymol (The PyMOL Molecular Graphics System, Version 2.0 Schrödinger, LLC). Atomic coordinates and structure factors have been deposited in the protein data bank under the accession codes 6sat, 6scp, 6scq, and 6scs.

**FtsZ and SepF polymerization assay.** Purified FtsZ and SepF were precleared at 25,000 × g for 15 min at 4 °C. FtsZ with or without SepF was added to a final concentration of 15 μM each in 100–500 μl final volume in polymerization buffer (100 mM KCl, 10 mM MgCl$_2$, 25 mM Pipes pH 6.9). The mixture was placed into a quartz cuvette with a light path of 10 mm or a 96-well plate and 0.5−2 mM GTP or GMPCPP were added to the reaction mixture. Data acquisition started immediately using an UV–Visible Spectrophotometer (Thermo scientific Evolution 220) during 600 s at 25 °C using 400 nm for excitation and emission and spectra with slits widths of 1 nm or a Varioskan™ LUX plate reader (Thermo Fisher Scientific) with an excitation of 400 nm. Experiments were done in triplicates and measurements were taken every 15 s during 600 s and keeping a constant temperature of 25 °C.

To follow the polymerization of SepF in the presence of lipids, we used SepF or SepF mutants plus SUVs at a final concentration of 50 μM each in polymerization buffer. The mixture was placed into a quartz cuvette with a light path of 10 mm and data acquisition was carried out as mentioned above. When used, FtsZ$_{CTD}$ was added at a final concentration of 100 μM.

**Small unilamellar vesicle (SUV) preparation.** Reverse phase evaporation was used to prepare small unilamellar vesicles (SUVs). A 10 mM lipid solution made of an 8:2 mixture of 1-palmitoyl-2-oleoyl phosphatidylcholine (POPC) and 1-palmitoyl-2-oleoylglycero-3-phosphoglycerol (POPG) (Avanti Polar Lipids) was prepared as described[68]. Chloroform was removed by evaporation under vacuum conditions and the dried phospholipid film was resuspended in a mixture of diethyl ether and buffer (25 mM Hepes pH 7.4, 10 mM MgCl$_2$ and 150 mM KCl). Diethyl ether was eliminated by reverse phase evaporation and by slowly decreasing the pressure to the vacuum. SUVs were obtained by sonication during 30 min at 4 °C. The diameter and charge of SUV were determined by measuring dynamic light scattering (DLS) and electrophoretic mobility profiles on a Zetasizer Nano instrument (Malvern Instruments).

**Real-time surface plasmon resonance (SPR).** All experiments were carried out on a Biacore T200 instrument (GE Healthcare Life Sciences) equilibrated at 25 °C in 25 mM HEPES pH 8, 150 mM KCl, 0.1 mM EDTA. For SPR surface preparation His6-Sumo tagged constructs of SepF$_{\Delta ML}$, SepF$_{\Delta ML,F131A}$, SepF$_{\Delta ML,K125E/F131A}$ were covalently immobilized on 3 independent flow-cells of an NTA sensorchips (GE Healthcare Life Sciences) as previously described[69]. The final immobilization densities of SepF variants ranged from 3000 to 4000 resonance units (RU; 1RU ≈ 1 pg per mm$^2$).

For SPR-binding assays, different concentrations of the FtsZ$_{CTD}$ peptide (ranging from 1.56 to 400 μM) were injected sequentially at 50 μl/min on the SepF-functionalized surfaces. Association was monitored for 30 s, followed by a buffer wash for 120 s during which the full dissociation of the SepF–FtsZ$_{CTD}$ complex was observed. The concentration dependence of the SPR steady-state signals ($R_{eq}$) was analyzed, allowing to determine the equilibrium dissociation constant Kd, by fitting the dose/response curve with the equation $R_{eq} = R_{max} * C/Kd + C$ (where $C$ is the concentration of FtsZ$_{CTD}$ and $R_{max}$ the response at infinite peptide concentration).

**Phase contrast and fluorescence microscopy.** For imaging, cultures of *C. glutamicum* were grown in BHI or minimal medium CGXII during the day, and washed and inoculated into CGXII media for overnight growth. The following day cultures were diluted to OD$_{600}$ = 1 and grown to the required OD (early exponential phase) for imaging. For HADA labeling, cultures were incubated with 0.5 mM HADA for 20 min at 30 °C in the dark. Two percent agarose pads were prepared with the corresponding growth medium and cells were visualized using a Zeiss Axio Observer Z1 microscope fitted with an Orca Flash 4 V2 sCMOS camera (Hamamatsu), and a Pln-Apo 63×/1.4 oil Ph3 objective. Images were collected using Zen Blue 2.6 (Zeiss) and analyzed using the software Fiji[70] and the plugin

MicrobeJ[71] to generate violin plots and fluorescent intensity heatmaps. Statistical analyses of cell lengths are summarized in Supplementary Tables 5 and 6. The experiments were performed as biological triplicates. Some autofluorescence is observed for wild-type *C. glutamicum* as shown in Supplementary Fig. 24.

**Western blots.** To prepare cell extracts, bacterial cell pellets were resuspended in lysis buffer (50 mM Bis-Tris pH 7.4; 75 mM 6-Aminocaproic Acid; 1 mM MgSO4; Benzonase and protease Inhibitor), and disrupted at 4 °C with 0.1 mm glass beads and using a PRECELLYS 24 homogenizer. Crude extracts (120 μg) were analyzed by SDS-PAGE, electro-transferred onto a 0.2 μm Nitrocellulose membrane and blocked with 5% (w/v) skimmed milk. Membranes were incubated with an anti-SepF antibody (produced by Covalab, Supplementary Fig. 25a), an anti-FtsZ antibody (produced by Proteogenix, Supplementary Fig. 25b) or an anti-mNeonGreen antibody (purchased from Chromotek) for 1 h at room temperature. After washing in TBS-Tween buffer (Tris-HCl pH 8 10 mM; NaCl 150 mM; Tween20 0,05% (vol/vol), the membrane was incubated with an anti-rabbit horseradish peroxidase-linked antiserum (GE healthcare) for 45 min. The membrane was washed and revealed with HRP substrate (Immobilon Forte, Millipore) and imaged using the ChemiDoc MP Imaging System (BIORAD). All uncropped blots are shown in Supplementary Fig. 26.

**SepF and FtsZ quantification.** In order to quantify the amount of SepF and FtsZ in the cell, purified recombinant SepF and FtsZ were serially diluted from 50 ng to 1.65 ng and loaded in a 12% gel. Proteins were electro-transferred onto a nitrocellulose membrane and immunodetected with the corresponding antibody. Quantification of the bands was performed using the software Image Lab (Biorad) and the band volume was plotted against the ng loaded in order to obtain a strand curve. *C. glutamicum* ATCC13032 was grown in CGXII media with 4% sucrose or 4% sucrose + 1% gluconate. Two time points were harvested at 6 h and 24 h (ON) and cell pellets were prepared as described above. 60 μg of the whole-cell extracts were loaded in a 12% gel and electro-transferred onto a nitrocellulose membrane. SepF and FtsZ antibodies were used to detect the proteins in the cell extracts and quantified using the volume tool in the Image Lab software.

**Electron microscopy.** For negative stain sample preparations, incubations were performed at room temperature. SUVs and SepF constructs (50 μM) were incubated in polymerization buffer (100 mM KCl, 10 mM MgCl$_2$, 25 mM Pipes pH 6.9) for 10 min. When used, FtsZ$_{CTD}$ was added at a final concentration of 100 μM. To image the FtsZ polymers with and without SepF, we used a final concentration of protein at 21 μM and 13 μM, respectively, in EM buffer containing 50 mM HEPES pH 7.4, 300 mM KCl, and 10 mM MgCl$_2$ supplemented with 3 mM GTP. FtsZ was incubated with or without SepF at room temperature and imaged after 10 min.

To image SepF rings form *M. tuberculosis*, *Mtb*SepF$_{\Delta ML}$ was used at a final concentration of 50 μM in buffer containing 50 mM HEPES pH 7.4, 300 mM KCl and 10 mM MgCl$_2$. For all samples, 400 mesh carbon coated grids (Electron Microscopy Sciences; CF 400-Cu) were glow-discharged on an ELMO system for 30 sec at 2 mA. 5 μl samples were applied onto the grids and incubated for 30 s, the sample was blotted, washed in 3 drops of water and then stained with 2% (weight/vol) uranyl acetate. Images were recorded on a Gatan UltraScan4000 charged coupled device (CCD) camera (Gatan) on a Tecnai T12 BioTWINLaB6 electron microscope operating at a voltage of 120 kV.

For cryo-EM, the ternary complex was prepared by incubating 100 μM SUVs, 10 μM SepF and 10 μM FtsZ in polymerization buffer (100 mM KCl, 10 mM MgCl$_2$, 25 mM Pipes pH 6.9; 3 mM GTP) for 10 min at room temperature. 5 μl of sample were deposited onto glow-discharged lacey carbon copper grids (Ted Pella) and plunge-frozen in liquid ethane using a Leica EM-CPC. Cryo-EM data acquisition was performed on a JEOL 2200FS (Jeol, Japan) 200 kV cryo-electron microscope equipped with an Omega in-column energy filter. High magnification (30,000×, corresponding pixel size 0.32 nm) zero-loss (slit: 20 eV) images were collected at nominal defocus between 1 and 4 μm depending on the experiment on a Gatan USC1000 slow scan CCD camera.

**Mass spectrometry.** Strains expressing Scarlet, SepF-Scarlet, and SepF$_{K125E/F131A}$-Scarlet were grown in CGXII minimal media supplemented with 4% sucrose and 1% gluconate for 6 h at 30 °C. Cells were harvested, washed, and normalized by resuspending cell pellets in PBS-T (1X PBS, 0.1% v/v Tween-80) to give a final OD$_{600}$ of 3. The cell suspensions were cross-linked with 0.25% v/v of formaldehyde for 20 min at 30 °C with gentle agitation. The crosslinking reaction was stopped by adding 1.25 M glycine and incubated for 5 min at room temperature. Cells were resuspended in 50 mM Bis-Tris pH 7.4, 75 mM aminocaproic acid, 1 mM MgSO4, 1× Benzonase, 1× Complete Protease inhibitors cocktail (ROCHE) and disrupted at 4 °C with 0.1 mm glass beads and using a PRECELLYS 24 homogenizer. After lysis, 0.1% n-Dodecyl-β-D-Maltopyranoside was added and the total extract was incubated for 1 h at 4 °C. Cell lysates were centrifuged at 4 °C for 10 min at 14,000 rpm and the soluble fraction was collected and protein levels quantified.

Detergent-solubilized protein extracts were incubated with 50 μl of magnetic beads cross-linked with 10 μg of purified anti-Scarlet antibody (produced by Covalab). Beads were collected with a magnetic stand and washed three times with lysis buffer and bound material was eluted in 0.1 M glycine pH 2.5 for 10 min at

4 °C, and immediately neutralized by adding 1 M Tris pH 9. The protein samples were reduced, denatured (2 M Urea) and alkylated with iodoacetamide prior to treatment with trypsin. Tryptic digests were cleaned on a POROS$^{TM}$ R2 resin (Thermo Fisher), vacuum dried and resuspended in phase A buffer (0.1% formic acid). Tryptic peptides from three replicates of each condition were analyzed using a nano-HPLC (Ultimate 3000, Thermo) coupled to a hybrid quadrupole-orbitrap mass spectrometer (Q-Exactive Plus, Thermo). Peptide mixtures were separated on a C18 column (PepMap® RSLC, 0.075 × 500 mm, 2 μm, 100 Å) using a 65 min gradient of mobile phase B from 0 to 55% (A: 0.1% formic acid; B: 0.1% formic acid in acetonitrile). Online MS analysis was carried out in a data-dependent mode (MS followed by MS/MS of the top 12 ions) using a dynamic exclusion list. PatternLab for Proteomics software[72] was used for protein identification against a target-reverse *C. glutamicum* database (Uniprot November 2018) to which the sequences of Scarlet, SepF-Scarlet and SepF$_{K125E/F131A}$-Scarlet were added.

Patternlab for proteomics was used for label-free quantitation analyses using extracted ion chromatogram (XIC). To calculate fold enrichment of FtsZ in pull down analyses, FtsZ signals (ΣXIC signal of detected peptides in each replicate) were normalized by comparing the ratio FtsZ/SepF in each case (for that we considered the intensity of SepF as (SepF-Scarlet minus Scarlet) or (SepF$_{K125E/F131A}$-Scarlet minus Scarlet)). The results are shown in Supplementary Table 2.

**Dynamic light scattering**. Before DLS experiments, protein samples were centrifuged at 25,000 × $g$ for 15 min at 4 °C. SUVs with or without SepF (or SepF mutants) were used at a final concentration of 50 μM each in 100 mM KCl, 10 mM MgCl$_2$, and 25 mM Pipes pH 6.9 Buffer and the reaction was carried out for 10 min at room temperature. Polydispersity of the samples was measured using Dyna-pro Plate Reader (Wyatt technology) equipped with an 830 nm laser and a temperature control module. The *Dynamics* software (version 7.9) was used to schedule data acquisition and data analysis. For each well, 20 measurements of 10 s were averaged and this operation was repeated 20 times for each condition at 25 °C. For samples containing FtsZ$_{CTD}$, the peptide was used at a final concentration of 100 μM. All the experiments were carried out as minimum three biological and technical replicates.

**Analytical ultracentrifugation (AUC)**. Protein samples were desalted using a PD Minitrap G-25 column (Sigma-Aldrich) into AUC buffer (50 mM Hepes pH 7.4, 150 mM KCl, 10 mM MgCl2) and analyzed at 1–2 mg/ml concentration. Samples were centrifuged at 42,000 rpm in a Beckman Coulter XL-1 analytical ultracentrifuge at 20 °C in a four-hole AN 60–Ti rotor equipped with 12-mm double-sector epoxy centerpieces. Detection of SepF concentration as a function of radial position and time was performed by optical density measurements at 280 nm and interferometry. Data analysis for sedimentation velocity was performed by continuous size distribution c(s) using Sedfit software version 15.01.

**GTPase activity assay**. The FtsZ GTPase activity was measured by detection of free phosphate using Biomol Green (Enzo Life Science). F15μM of SepF and 15 μM of FtsZ with or without 15 μM of SepF were incubated at 4 °C for 5 min in polymerization buffer (100 mM KCl, 10 mM MgCl2, 25 mM Pipes pH 6.9). After, 100 μl of the mixture were place in a 96-well-plate that contained GTP at 2 mM final concentration. Reactions were incubated at room temperature for 10 min and 100 μl of the developing reagent was added. The mixture was incubated at room temperature for 25 min to allow development of the green color and absorbance was measured at 620 nm using a Varioskan™ LUX (Thermo Fisher Scientific). Reactions were performed in triplicate for each condition. Obtained values were normalized by the lowest SepF value and free phosphate was measured by comparison to a phosphate standard curve.

**Far-UV circular dichroism**. The secondary structure of SepF$_M$ in the presence or absence of SUVs was determined using synchrotron radiation circular dichroism (SRCD) carried out at the beamline DISCO (SOLEIL, Gif-sur-Yvette, France). Three individual scans were averaged to obtain final far-UV spectra. Measurements were made at 25 °C with an integration time of 1.2 s and bandwidth of 1 nm. SRCD spectra in the far-UV (190 to 250 nm) were recorded using QS cells (Hellma, France) with a path length of 100 μm. SepF$_M$ was used at 100 μM in CD buffer (25 mM Hepes pH 7.4, 100 mM KCl) in the presence and absence of SUVs (POPC: POPG 8:2) at a final concentration of 2 mM. As a blank, the CD buffer was used in the absence or presence of SUVs, and subtracted from far-UV CD spectra. Finally, BestSel[73] was used in order to estimate the content of secondary structure.

**Lipid peptide interaction (tryptophan fluorescence emission titration)**. To estimate the partition coefficient ($K_x$) between SUVs and SepF, we used the synthetic peptide MSMLKKTKEFFGLAW (purchased from Genosphere), which contains the SepF$_M$ sequence with an extra W residue at the C-terminal. $K_x$ is defined as the ratio of peptide concentration in the lipid and in the buffer phases. $K_x$ can be expressed by the following equation:

$$K_X = \frac{P_L/(P_L + [L])}{P_W/(P_W + [W])}$$

in which $P_W$ represents the concentration of soluble peptide (in aqueous phase) and $P_L$ the peptide concentration bound to lipid membranes (lipidic phase). [L] refers to the lipid concentration and [W] refers to the water concentration. $K_x$ is directly related to the apparent dissociation constant as $K_x * Kd = [W]$ with $Kd * P_L = P_W * [L]$. The KaleidaGraph software was used to fit the $K_x$ to the experimental data (for details, see ref. [68]).

We used a FP-8200 (Jasco, Tokyo, Japan) spectrophotometer equipped with a thermostatic Peltier ETC-272T at 25 °C. Experiments were performed in a cuvette 109.004F-QS (Hellma, France) containing a magnetic stirrer. A bandwidth of 5 nm was used for emission and excitation. One micromolar of peptide was used in Titration buffer (100 mM KCl and 25 mM Hepes pH 7.4). We measured the florescence emission between 300 and 400 nm at a scan rate of 125 nm/min with an excitation wavelength of 280 nm. The obtained spectra were corrected by blank subtraction (SUV light scattering in titration buffer). Next, the maximum wavelength value ($\lambda_{max}$) and the fluorescence ratio 330/370 were calculated to measure the partition coefficient ($K_x$).

**Phylogenetic analysis**. We performed sequence homology searches with HMMER[74] using SEPF_BACSU/ Q8NNN6_CORGL as queries against the UniprotKB database. More than 4000 sequences were then clustered at 76% with CD-HIT, and after removal of short (<120 residues) and long (>240) candidates, 1800 sequences were kept. Redundancy was further reduced at 50% sequence identity with clustering programs CD-HIT[75] and MMseqs2[76] separately, merging both outputs and removing duplicate sequences. We used this non-redundant dataset of 839 sequences with wide taxonomic distribution to compute a multiple alignment with MAFFT[77] (l-insi option). Before phylogenetic reconstruction, we used trimAL[78] to remove columns with >80% gaps. Finally, we built the SepF tree with PhyML 3.3[79] using the LG+CAT substitution model, chosen with SMS[80]. Trees were visualized using FigTree v1.4.3.

**Reporting summary**. Further information on research design is available in the Nature Research Reporting Summary linked to this article.

## Data availability

The crystallographic data is available from the Protein Data Bank (www.rcsb.org), under the accession numbers (PDB codes) reported in Supplementary Table 1. The source data underlying Figs. 1b, d, 2e, 3a, 3b, and 5f and Supplementary Figs 1a, 1c, 2b, 4c, 5a, 7c, 7d, 11b, 16, 19b, 20b, 20d, and 21 are provided as a Source Data file. All other data are available from the corresponding authors upon request.

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

## Acknowledgements

We thank A. Ducret for help with MicrobeJ, F. Gubellini for help with electron microscopy, M. Bott and M. Baumgart for the pk19-P3323-*lcpA* plasmid and help with corynebacterial genetics, and H. Gramajo for the pTGR5 plasmid. We gratefully acknowledge the core facilities at the Institut Pasteur C2RT, in particular G. Pehau-Arnaudet (UBI), B. Raynal, S. Brule (PFBMI), P. Weber, C. Pissis (PFC), and J. Fernandes (UtechS PBI/Imagopole, supported by France BioImaging; ANR-10–INSB–04; Investments for the Future). We thank the staff of ESRF and of EMBL-Grenoble for assistance and support in using beamlines ID30B and ID23-1, and the staff of SOLEIL Synchrotron for assistance in using the beamline Disco. We acknowledge the PICT-IBISA for providing access to the cryo-EM facility at Orsay. Finally, we would like to thank the reviewers for their coments and suggestions, which have helped us to improve the quality of the manuscript. This work was partially supported by grants from the Institut Pasteur (Paris), the CNRS (France) and the Agence Nationale de la Recherche (PhoCellDiv, ANR-18-CE11-0017-01). A.S. is part of the Pasteur-Paris University (PPU) International Ph.D Program, funded by the European Union's Horizon 2020 research and innovation program under the Marie Sklodowska-Curie grant agreement No 665807. Q.G. was funded by MTCI Ph.D school (ED 563); A.V. was supported by a DIM MalInf (infectious diseases) grant. M.G. acknowledges support from Programa de Desarrollo de las Ciencias Básicas and Sistema Nacional de Investigación e Innovación, Uruguay.

## Author contributions

A.M.W. and P.M.A. designed the research. A.S., M.M., Q.G., M.B.A. and A.M.W. conducted the protein biochemistry, cell biology and genetic experiments, and purified proteins for structural and biophysical studies. P.E. and A.S. carried out the biochemical and biophysical studies of protein–protein interactions. A.V, A.C., and A.S. carried out binding studies of lipid membrane-protein interactions. A.M.W. and R.D. carried out MS and proteomic experiments. A.S., A.H., A.M.W., and P.M.A. carried out the crystallogenesis and crystallographic studies. M.G. and A.M.W. performed the phylogeny analyses. M.V.N. contributed essential reagents (fluorescent labeled amino acids). A.S., S.T., and A.M.W. performed the cryo-EM and negative stain EM studies. A.M.W. and P.M.A. wrote the paper. All authors edited the paper.

## Competing interests

The authors declare no competing interests.
