## [Peer Review File · Nature Communications]

Reviewers' comments:

Reviewer #1 (Remarks to the Author):

Actinobacteria lacks the canonical FtsZ-membrane anchors such as FtsA and ZipA, instead it has SepF. Here, the authors mainly describes the essential role of SepF in *C. glutamicum*, the high-resolution crystal structure of the SepF C-terminal core domain in complex with the C-terminal peptide of FtsZ, and SepF and FtsZ oligomerization upon lipid membrane-binding. Based on these biochemical studies, the author propose the model of the role of dynamic SepF oligomerization. The manuscript is definitely recommendable for publication, in my opinion. However, there are clearly missing data and description which may be misleading. My critical concerns are as follows:

Comments/questions:

1) Page 7, top

The author determined the structures of the SepF in complex with the C-terminal peptide of FtsZ as well as SepF only. Is there any induced fit upon the peptide binding?

2) Page 7, line 10

"This 2:2 binding stoichiometry can explain ... (Fig. 2e)"

The author measured the absorbance by changing the ratio of SepF and FtsZ, but the stoichiometric SepF:FtsZ ratios in Fig. 2e are only 1:1, 1:50, and 0:1. The data is not enough to prove the 2: 2 binding stoichiometry. How about 1: 2 or 2: 1 stoichiometry in this measurement?.

3) Page 7, line 20

"the electron density map (Supplementary Fig. 8a)"

The author should define the electron density map coefficients. (e.g., 2fo-fc)

4) Page 9, line 15

"A putative regulatory role for the C-terminal helix of SepF"

The discussion on the C-terminal helix switch is interesting, but there are many uncertainties because no regulatory molecules have been identified. In the abstract, the helix switch is also emphasized. I think it seems better to express weakly.

5) Page 10, bottom & Supplementary Fig 12

"differs from the parallel orientation seen in the *B. subtilis* SepF structure, thus pointing to species-specific polymerization mechanisms"

Species-specific polymerization mechanisms are unlikely. In Supplementary Fig 12d, PDB code 3P04 was chosen, but 3P04 seems to be a hypothetical protein. I guess 3P04 is difficult to compare with SepF. In addition, protein molecules in the biological polymers should be in a translational relationship in general. Also, it seems to be inconsistent with the model shown in Fig. 6. Require further explanation.

Reviewer #2 (Remarks to the Author):

In this manuscript, the authors demonstrate that SepF is essential in *Corynebacterium glutamicum* and important for recruiting FtsZ to the mid-cell and for assembling a functional Z-ring. SepF interacts with lipid membranes through the predicted helix at the terminus, tubulates lipids and has a bundling effect on FtsZ protofilaments. The authors also crystallized the SepF C-terminal core domain in complex with a peptide comprising the FtsZ C-terminal domain. The current study suggests that the helix $\alpha 3$ plays a regulatory role on SepF polymerization by uncovering the outer face of the β -sheet for intermolecular interactions. The study also demonstrates that SepF oligomerization occurs upon membrane binding and, interestingly, is reversed by FtsZ interactions. SepF polymerization has a membrane remodeling effect, antagonized by FtsZ binding, where the

authors suggest that this might aid in septum formation during division and serve as a checkpoint. This is a well written manuscript that makes an important contribution to understanding the functional roles of FtsZ regulators during division in Actinobacteria. My comments are as follows:

1. Line numbers would be helpful for review purposes.
2. Fig. 1d and 1f – axis labels are unreadable
3. Fig. 1c and descriptive results (page 5). Although lysis is discussed as occurring “eventually” after 3 hours, branching is described at 6 and 9 hours. The descriptive timeline is confusing. What is the frequency and timing of the lysis events since SepF appears gone by western blot after 2h. What is the phenotype upon overexpression of SepF, does this condition mislocalize Z-rings or increase the number of rings per cell? What are the relative concentrations of both proteins in vivo under wild type conditions?
4. Fig 2f – Several controls have been omitted from the electron micrographs shown but are very important for interpretation of the result, including FtsZ+SepF (no GTP), SepF alone, FtsZ (no GTP). The FtsZ+GTP is very difficult to see. What are the polymer widths +/- SepF?
5. Fig 2g – The authors should show several concentrations on the SPR for each SepF protein in Fig. 2. Since they are each immobilized on different areas, there is not much information to be gained from a single CTD concentration RU comparison, which can be significantly impacted by mass effects and relative activities after surface coupling. This is especially true for weak binding events and fast off rates. Are Kds for full length FtsZ binding (not the CTD) to SepF similar to or stronger than CTD, and different with and without GTP? It is expected that the Kds would be strongest for full length with GTP.
6. How does hydrolysis by FtsZ regulate the interaction with SepF? Do stable polymers, such as those stabilized by GMPCPP, interact with SepF similarly as GTP? Does the SepF-lipid destabilization function require the FtsZ GTP hydrolysis cycle under conditions that are closer to physiological concentrations? In assays with peptide alone, this cannot be addressed because the peptide may be present at high concentration than the protein in vivo, and the binding region on the peptide may be restricted or more accessible in various FtsZ conformation and not presented as in polymerized FtsZ.
7. Add labels to lines in Fig. 1b, 3a and violin clusters in 3b.
8. In complementation assays with SepF-scarlet wt and mutants, was lysis observed?
9. Does SepF modify FtsZ GTP hydrolysis rates in vitro? The dynamic bundling model proposed here is reminiscent of an interaction between FtsZ and the assembly regulator GpsB in *Staphylococcus aureus*, which bundles but then accelerates disassembly via stimulation of GTP hydrolysis, also serving a checkpoint-like role in vivo (Eswara, et al., 2018). GpsB, however, functions in contrast to most bundling proteins which are largely thought to slow down dynamics or have no effect on FtsZ-GTP turnover; therefore it is important to understand how SepF affects FtsZ hydrolysis and dynamics.

Reviewer #3 (Remarks to the Author):

Summary:

The manuscript presented here is a study of the physiological function of *Corynebacterium glutamicum*'s SepF protein. SepF is conserved in many Gram positive bacteria and has been

previously shown to interact directly with FtsZCTD, and to have a role in cell division. This work is supported by many biochemical and in vivo approaches. In this organism that lack homologs of other well studied FtsZ binding proteins and regulators of the Z ring formation understanding the role of SepF in these processes is critical.

Many of the experiments (especially all the biochemistry) presented in that study are performed well, with the adequate control and presented with enough details that the reader can be confident in the results. These results then support many of the authors conclusions.

The major issues I have with the manuscript concern the experiments in which fusions to fluorescent proteins are used to localize different variants of SepF and FtsZ in vivo. As I detailed below I do think the authors need to improve the experiments presented here and add a few controls to make the reader confident that these experiments supports the authors conclusions. Because of the lack of confidence on the results obtained in these in vivo experiments, I do not believe the model presented at the end of the manuscript is accurate or well supported.

- Part about SepF being essential:

The authors do show convincingly that depletion of SepF impairs cell division (no formation of new septa and of FtsZ rings) but does not block peptidoglycan (PG) synthesis at the poles, leading to some filamentation and then branching of the depleted cells.

I would like the authors to propose an explanation of why branching might occur along the long axis of the cells? How depletion of SepF will lead (after about 6h according to Supl. Fig 2) to new PG synthesis (shown by HADA labelling) on the side of the bacteria, which then seem to lead to the formation of branches. Why is PG synthesis not restricted at the poles even after 6h since no septa are formed?

When the authors claims that mNeon-FtsZ represent about 3% of the total levels of FtsZ [Bottom of page 5] are they talking of the conditions were there is leakage from the promoter when the cells are grown in sucrose, am I correct? I think the authors should add the fact that the cells Fig 1e or Supl. Fig 5 are also grown in presence of 4% sucrose (I know it is said into the material and methods but it would simplify reading the manuscript if it was clear in the figure legends too).

Can the author check that the expression of mNeon-FtsZ is not affected by the presence of 1% myo-inositol, adding a lane in their western Supplemental Fig4b? My concern is with the pictures presented Fig 1e in which the background level of green fluorescence seems very high and the cells have a lot of cytoplasmic fluorescence not localized into a ring, even a t=0. Is this due to a high expression of the mNeon-FtsZ or a protease processing of the fusion protein leading to a lot of mNeon not fused to FtsZ in the cytoplasm? It is important to know if the level of FtsZ is somehow increased as it may also participate in the inhibition of division phenotype.

Finally, the mNeon-FtsZ, a t=3 in Fig 1e appears to form some kinds of polymers in the cytoplasm of the cells depleted for SepF, but with the green fluo background it is kind of hard to see. Can the authors comments on this? This would suggest that FtsZ at least in vivo could form some kinds of structures (may be bundles) in absence or in condition of depletion of the membrane anchor? This can be important for the authors discussion when they present their model on how SepF may regulate FtsZ bundling.

- Part about Molecular details of SepF interactions:

-SepF binds to the membrane through its N terminal amphiphatic helix linked to the core of the protein through a long disordered linker. The authors do show that the N terminal first 14 amino acids peptide of SepF sequence that is predicted to form and amphiphatic helix can interact with lipids in vitro (Supl Fig 6) But, a convincing appropriate experiment to show that this sequence is a real membrane targeting sequence in vivo would have been to fuse at the C-terminal of that helix followed by the disordered linker of SepF, a fluorescent protein (GFP, MNeonGFP or Scarlet) and show that this helix can target the fluorescent proteins to the membrane of *C. glutamicum* or even

in E coli. Such an experiment would strengthen the authors conclusions.

-The authors do show that *in vitro*, even the SepF Δ MML unable to bind to the membrane can still interact with FtsZ to bundle FtsZ filaments but that when they mutate 2 residues in SepF that are important in the FtsZCTD binding according to their structure work, SepF loose that ability. Following these experiments, the authors looked into the *in vivo* properties of these proteins and fused them to the Scarlet fluorescent protein. One comment about this is that I assume the Scarlet protein is fused on the C-terminal side of SepF as this is not described properly in the manuscript. It would help the reader if the authors would at least add in their diagrams of the different SepF species in Fig 3c the position of the linker and Scarlet protein they added to SepF.

I do have many concerns about these SepF-Scarlet fusions, that make me think the fusion protein is not behaving as expected. First the authors say themselves that SepF-Scarlet fusion only partially complemented the SepF depletion strain. Because the cells expressing it are elongated compared to the control cells (Fig 3 and Sup. Fig 11). When I observe the level of expression of the fusion (Sup Fig 11), I am worried that the observed increase in length of the cells is due of the very high level of expression of the fusion instead of its inability to complement the depletion of WT SepF. The authors should definitively show that similar expression of WT SepF (not fused to any tags) are not toxic for the cells and do not cause an increase in size. Also the Westerns presented Sup Fig 11, show that in addition of the very high levels of expression of these SepF-Scarlet fusion it seems like there is also a lot of proteolysis of the fusion (multiple bands on the lanes when detected with the SepF antibody). The authors should at least do a similar western using an antibody against Scarlet to see what is the actual fluorescent species we are looking at in these cells. I am very concerned because in the cells at T=0 where the authors show sup Fig 11 there is no detectable induction of the fusions proteins by western, there is already enough fluorescence to see localization at the septa of all the proteins fusions (Fig 3c). Then at 6 hours, the fluorescence observed in the cells does not seem to be increased by a lot compared to the t=0 but on the western the expression of the fusion proteins is a lot higher, with multiple bands of different molecular weights (sup Fig 11). In the case of the SepF Δ MML, despite having the same fluorescence levels in the cells that for the other fusion, the expression level of the proteins measured by Western (Sup Fig 11) is not very much. Is that protein unstable, which could explain why it does not complement?

For the SepF mutants that do not bind FtsZCTD there is so much cytoplasmic fluorescence that I do not think the authors could see if there is any membrane localization. May be a proteolytic version of the fusion (the band at 25KDa) for example is causing so much cytoplasmic fluorescence that it would make impossible to determine if the fusion that is still full length would bind to the membrane. The results with the SepF-Scarlet fusion of the mutant unable to bind to FtsZ are inconsistent with the *in vitro* results the authors talk about later in the manuscript (Fig 5). *In vitro* the authors claim that SepF not bound to FtsZ can attach to the lipid membranes (it makes the SUV tubulated) but then *in vivo* it does not seem to be the case as their mutant is cytoplasmic. I do think the authors should address these differences. May be the fusion on the C-terminal of SepF blocks its ability to polymerize and this could reduce its ability to bind to the membranes (in the literature E coli's MinD seems to need to be a polymer to be able to bind to the membrane as it was shown that the amphiphatic helix of E coli's MinD is too short to direct a monomer to the membrane). If this is the case for SepF then this could change the interpretation of the results to propose a role of SepF in allowing Z ring formation.

Finally the fact that all these fusion proteins are able to be detected at the septum at T=0 in the pictures Fig 3c, strongly suggests that these proteins are still able to be associated with the wildtype SepF (before it is depleted) and despite not being able to localize to the membrane or to interact with FtsZ, these proteins still do not seem to cause major problems at the division site because all the cell size of a t=0 look similar with all the 3 constructs tested.

I do think these experiments with the fusion to scarlet needs to be re addressed before publication as I do not think these confidently support the conclusions the authors are claiming. I would suggest to not even induce the expression of the scarlet fusions as the leak from the gntK promoter is enough to observe the fluorescence (as demonstrated a t=0 in Fig 3c). An alternative

would be to perform immunofluorescence experiments using the anti SepF antibody (after affinity purification) instead of using the fusion to Scarlet.

- Part about the role of the C terminal helix of SepF.

-The authors argue that the structures of SepF unliganded vs SepF bound to the FtsZCTD are the same overall (bottom of page 9). Based on previous structure analysis of other SepF homologs the authors suggest that this C terminal helix $\alpha 3$ might be capping another domain of SepF involved in the polymerization of SepF homodimers in solution. But, the only situations where the authors can observe this type of interaction is in the crystal structure of a SepF deleted for that $\alpha 3$ helix or when it is associated with lipid vesicles. The authors also show that the binding of SepF to the FtsZ CTD reverse that polymerization ability onto lipid membranes. As pointed out above, there are contradictory results between the experiments done in vivo using the fusion with Scarlet (Figure 3) and these in vitro experiments. In vivo SepF does not seem to be bound to the membrane when it is not interacting with the CTD of FtsZ while in vitro it seems to be able to bind and polymerize at the lipid membrane when it unbound to FtsZ. The authors should discuss this.

-Finally the authors try to argue that forming heterodimer of WT SepF with the SepF mutated for its interaction with FtsZ and fused to Scarlet reduce the interaction of a SepF polymer with FtsZ which lead to formation of increased SepF fluorescent rings into the cells when compared to the same experiment using the WT SepF fused to Scarlet as a control. Since as pointed out above the behavior of these scarlet protein does not seem to be well characterized, I think the authors should at least do the following additional control: according to their reasoning if reducing the interaction of SepF with FtsZ lead to a better polymerization of SepF at the membrane to form numerous rings in the cells then expression of wild type SepF-Scarlet into conditions of FtsZ depletion should have the same effect an also leads to many SepF rings in the cells. Such a result would support their conclusions.

-Another note here to about these Scarlet fusion proteins: According to Sup Fig 18 it looks like the mutant SepF unable to bind to FtsZ and fused to Scarlet is overexpressed a lot more that the WT SepF fused to Scarlet. In addition, that later protein is also processed proteolytically and give a strong additional band at above 37KDa (which is absent in the mutant lane). Such differences of overexpression could explain the fact that the WT SepF may makes less rings in the cells. Can the author comment on this?

- Model:

It is very hard to believe that FtsZ filament bundling (which increase the concentration of FtsZ molecules near the SepF molecules that also cause these filaments to be anchored at the membrane) can reduce access to FtsZCTDs. I would like the authors to propose a mechanism for this argument.

The authors propose in their model (page 14) that the FtsZ bundling and stabilization of FtsZ polymers would increase the local concentration of SepF at the membrane and start SepF polymerization. But they also say that SepF needs to bind FtsZ to go to the membrane (and reciprocally FtsZ binds to SepF to be attached at the membrane). They also say that SepF bound to FtsZ antagonize SepF polymerization, and they show that when an SepF molecule that cannot bind FtsZ is expressed causing a reduced ability of the WT SepF in the cells to binds FtsZ (probably by forming SepF heterodimers), leads to the formation of more SepF polymers (rings). But how does a reduce ability to bind FtsZ also promote recruitment of FtsZ filaments into bundles which is according to the authors model the cause of the start of SepF polymerization?

There are too many contradictory and confusing issues in the way the model is presented, and the authors should do a better job clarifying their model and indicate how it explains their experimental observations.

Reviewers' comments:

We would like to thank the reviewers for their positive comments about the general importance of the manuscript and their constructive criticism that we have now addressed by thoroughly revising the manuscript and by adding additional experiments in particular those required by referees 2 and 3. As asked by referee 2, and for easier reading of the changes carried out, we have added line numbers to the text. Our answers can be found point by point in the text below (in blue).

Reviewer #1 (Remarks to the Author):

Actinobacteria lacks the canonical FtsZ-membrane anchors such as FtsA and ZipA, instead it has SepF. Here, the authors mainly describes the essential role of SepF in *C. glutamicum*, the high-resolution crystal structure of the SepF C-terminal core domain in complex with the C-terminal peptide of FtsZ, and SepF and FtsZ oligomerization upon lipid membrane-binding. Based on these biochemical studies, the author propose the model of the role of dynamic SepF oligomerization. **The manuscript is definitely recommendable for publication, in my opinion. However, there are clearly missing data and description which may be misleading.** My critical concerns are as follows:

Comments/questions:

1) Page 7, top

The author determined the structures of the SepF in complex with the C-terminal peptide of FtsZ as well as SepF only. Is there any induced fit upon the peptide binding?

As indicated in the manuscript, from the structural superposition the overall rms between the unliganded and FtsZ-bound forms of SepF can be superimposed with an rms of 0.83 Å, suggesting that there is no induced fit upon peptide binding. This has now been explicitly included in the revised version (page 10, lines 12-15).

2) Page 7, line 10

“This 2:2 binding stoichiometry can explain ... (Fig. 2e)”

The author measured the absorbance by changing the ratio of SepF and FtsZ, but the stoichiometric SepF:FtsZ ratios in Fig. 2e are only 1:1, 1:50, and 0:1. The data is not enough to prove the 2: 2 binding stoichiometry. How about 1: 2 or 2: 1 stoichiometry in this measurement?.

The assessment of the 2:2 binding stoichiometry is not based on the absorbance measurements, but on the crystal structure of the complex, which clearly shows that a SepF dimer binds two copies of the FtsZ CTD peptide. The experiment shown in Figure 2 is meant to show that even at much lower levels of SepF (1:50, SepF:FtsZ) we see an effect on FtsZ dynamics and bundling). This in turn is important when thinking that in the cell the SepF levels are lower than those of FtsZ (see also answer to reviewer 2 point 3 as well as the new Supplementary Fig. 21).

3) Page 7, line 20

“the electron density map (Supplementary Fig. 8a)”

The author should define the electron density map coefficients. (e.g., 2fo- ρ)

This has now been indicated in the revised version (Supplementary Fig. 8, Legend).

4) Page 9, line 15

“A putative regulatory role for the C-terminal helix of SepF”

The discussion on the C-terminal helix switch is interesting, but there are many uncertainties because no regulatory molecules have been identified. In the abstract, the helix switch is also emphasized. I think it seems better to express weakly.

We agree with the reviewer that the hypothesis of a regulatory role for helix 3 remains speculative at this stage and needs to be confirmed by further experiments. We have therefore followed his/her advice to tone down this hypothesis in the revised manuscript, where it is now only mentioned in the main text (but not in the abstract) as a hypothesis, possibly regulated by yet-to-be-identified additional molecule(s) or posttranslational modifications (Page 11, lines 4-6 and Page 14, lines 4-6).

5) Page 10, bottom & Supplementary Fig 12

“differs from the parallel orientation seen in the *B. subtilis* SepF structure, thus pointing to species-specific polymerization mechanisms”

Species-specific polymerization mechanisms are unlikely. In Supplementary Fig 12d, PDB code 3P04 was chosen, but 3P04 seems to be a hypothetical protein. I guess 3P04 is difficult to compare with SepF. In addition, protein molecules in the biological polymers should be in a translational relationship in general. Also, it seems to be inconsistent with the model shown in Fig. 6. Require further explanation.

There is a misunderstanding here, 3P04 was chosen because it is the structure of *C. glutamicum* SepF (determined a few years ago by a structural genomics consortium and annotated at that time as hypothetical protein), i.e. the same protein we are dealing with in our manuscript. We have now changed the legend to old Suppl. Fig. 12 (now Suppl. Fig. 13 in the revised version) to make this clearer. Our main point here is that the same protein – *C. glutamicum* SepF – in three different crystal forms (namely 3P04 and our crystal structures of the C_{ter}-truncated form of the free and FtsZ_{CTD}-bound forms SepF) do show the same beta-beta interface, strongly suggesting that it could correspond to a polymerization interface. It is important to note that in the case of 3P04 the C-terminus was replaced by the affinity tag, which results in the same effect as removing helix 3 completely. Furthermore, this interaction is similar to the beta-beta interaction observed for *B. subtilis* SepF (Duman et al, PNAS, 2013), albeit with some differences that might be species-specific. As for the model shown in Figure 6, this has now been completely re-drawn (see new Figure 6) and rewritten (Page 14, Lines 12-28 and Page 15, lines 1-4). See also our answer to Reviewer 3, point 17, below.

Reviewer #2 (Remarks to the Author):

In this manuscript, the authors demonstrate that SepF is essential in *Corynebacterium glutamicum* and important for recruiting FtsZ to the mid-cell and for assembling a functional Z-ring. SepF interacts with lipid membranes through the predicted helix at the terminus, tubulates lipids and has a bundling effect on FtsZ protofilaments. The authors also crystallized the SepF C-terminal core domain in complex with a peptide comprising the FtsZ C-terminal domain. The current study suggests that the helix $\alpha 3$ plays a regulatory role on SepF polymerization by uncovering the outer face of the β -sheet for intermolecular interactions. The study also demonstrates that SepF oligomerization occurs upon membrane binding and, interestingly, is reversed by FtsZ interactions. SepF polymerization has a membrane remodeling effect, antagonized by FtsZ binding, where the authors suggest that this might aid in septum formation during division and serve as a checkpoint. **This as a well written manuscript that makes an important contribution to understanding the functional roles of FtsZ regulators during division in Actinobacteria.** My comments are as follows:

1. Line numbers would be helpful for review purposes.

This has been done in the revised version.

2. Fig. 1d and 1f – axis labels are unreadable

We have modified the labels in Figs 1d,1f to make them the same size as all other annotations.

3. Fig. 1c and descriptive results (page 5). Although lysis is discussed as occurring “eventually” after 3 hours, branching is described at 6 and 9 hours. The descriptive timeline is confusing. What is the frequency and timing of the lysis events since SepF appears gone by western blot after 2h.

We used the English definition of “eventually” meaning “in the end; ultimately”. To avoid confusion, we have now changed the text (Page 5, lines 5-6) to say that cells lyse in the end of the time course of the experiment. They do not lyse at timepoints 3 or 6 hours.

What is the phenotype upon overexpression of SepF, does this condition mislocalize Z-rings or increase the number of rings per cell?

To address this issue, we have made two new strains for ectopic expression of SepF (without a tag) under the control of the weak promoter PgntK and the strong promoter Ptac, respectively. Cells expressing SepF under the control of PgntK, which is a weak and tightly repressed promoter, either in induced or non-induced conditions, showed a wild-type like phenotype with normal division sites, validating our expression system for studying the localization of the different SepF mutants. On the other hand, when SepF is highly over-expressed under the control of Ptac, a strong promoter that is active in presence of IPTG, we observed very elongated cells. When the cells were stained with Nile red, we observed 1 or no septa. In some cells branching was observed. This is the same phenotype than the one described for *M. smegmatis* (Gola et al, Mol Microbiol, 2015). In the absence of IPTG inducer,

the cells have a wild-type like phenotype (the levels of pTac leakage are equivalent to those of maximal induction from the Pgntk promotor). We have now added a supplementary figure (new Supplementary Fig. 20) describing the phenotypes under the different promoters and the quantification of expression levels.

We also determined that SepF overexpression (from the Ptac promoter) mis-localized FtsZ-rings in the elongated cells (mNeon-FtsZ expressed from the Pgntk promoter in sucrose condition). FtsZ localizes in patches and cannot form Z-rings anymore. This shows the importance of the protein levels in the cell, as in the WT situation SepF is present 5-20 fold less than FtsZ (see the answer to the next point below). The new Suppl. Fig. 20 now includes the images and Western Blots; the new strains were included in the tables and materials and methods (Page 18, lines 3-7) and described in the main text (Page 13, lines 19-22).

What are the relative concentrations of both proteins in vivo under wild type conditions?

We have quantified by Western Blots the relative concentrations of SepF and FtsZ in wild type cells, grown in sucrose or sucrose plus gluconate in exponential (6h) and stationary (ON) phase. In exponential phase the ratio FtsZ:SepF is between 10-20 and decreases somewhat in stationary phase (around 8), showing that there is a variation (mostly through FtsZ levels depending of the growth phase and the source of carbon) but that FtsZ is always in large excess compared to SepF. Similar results were seen in *M. tuberculosis* where the ratio is about 5 (Schubert et al, 2015). These results are presented in the new Supplementary Fig. 21 and described in the main text (Page 14, lines 18-20), as well as in Materials and Methods (Page 24, lines 22-38 and Page 25, lines 1-5).

4. Fig 2f – Several controls have been omitted from the electron micrographs shown but are very important for interpretation of the result, including FtsZ+SepF (no GTP), SepF alone, FtsZ (no GTP). The FtsZ+GTP is very difficult to see. What are the polymer widths +/- SepF?

These controls had been done but were not included in the paper as they were not very informative (i.e. no filaments), representative images of the three conditions are shown below. However, we think they will not add useful visual information to the paper. Instead, we have now stated in the revised version (Page 7, lines 23-24) that there are no FtsZ filaments or bundles in the absence of GTP with or without SepF, nor for SepF alone. We have measured the polymer width and included this information in the legend of Fig. 2f (Page 37, lines 8-10). The polymers in the left-hand panel have a width of about 4nm which corresponds to the width of FtsZ. The bundles shown in the right-hand panel range in width from 20-60 nm.

FtsZ (no GTP)

FtsZ+SepF (no GTP)

SepF alone

5. Fig 2g – The authors should show several concentrations on the SPR for each SepF protein in Fig. 2. Since they are each immobilized on different areas, there is not much information to be gained from a single CTD concentration RU comparison, which can be significantly impacted by mass effects and relative activities after surface coupling. This is especially true for weak binding events and fast off rates. Are Kds for full length FtsZ binding (not the CTD) to SepF similar to or stronger than CTD, and different with and without GTP? It is expected that the Kds would be strongest for full length with GTP.

These experiments are indeed described in Supplementary Figure 9, which shows the binding of the different SepF forms at several concentrations of CTD in each case. We had decided to show only one curve for each mutant in the main Figure 2g as a comparative graph, as it is more visual in explaining the result. The Supplementary Fig. 9 is now referred to in the legend of main Fig. 2g.

We also attempted to carry out SPR experiments with full-length FtsZ plus GTP, but these failed probably due to protein polymerization.

6. How does hydrolysis by FtsZ regulate the interaction with SepF? Do stable polymers, such as those stabilized by GMPCPP, interact with SepF similarly as GTP?

We have performed the FtsZ polymerization assays in the presence of GMPCPP and SepF and observed a similar effect on bundling formation as that seen with GTP. These results suggest that GTP hydrolysis by FtsZ does not affect its interaction with SepF, as it could indeed be expected from the structural independence of the GTPase and CTD domains. These results have been reported in the main text (page 7, lines 24-26) and added to the existing Supplementary Fig. 7 (panels c & d) and Supplementary Materials and Methods.

Does the SepF-lipid destabilization function require the FtsZ GTP hydrolysis cycle under conditions that are closer to physiological concentrations? In assays with peptide alone, this cannot be addressed because the peptide may be present at high concentration than the protein in vivo, and the binding region on the peptide may be restricted or more accessible in various FtsZ conformation and not presented as in polymerized FtsZ.

A priori, the SepF-lipid destabilization function should not require GTP hydrolysis activity as the peptide alone is able to produce this effect. Nevertheless, we carried out SepF-lipid

polymerization assays in the presence of full-length FtsZ + GTP, but the results were not conclusive as we were unable to deconvolute the individual contributions of FtsZ and SepF polymerization from the total absorbance. The only way we have been able to look into the ternary complex involving both full-length proteins plus lipids and GTP was using cryo-EM (Figures 5c-d).

7. Add labels to lines in Fig. 1b, 3a and violin clusters in 3b.

Labels have now been added. In Fig. 3 the color codes are the same throughout the figure. To avoid excessive crowding, we have added labels only to the first panel, and included the complete descriptions in the legend.

8. In complementation assays with SepF-scarlet wt and mutants, was lysis observed?

In these strains no lysis was observed, as they complement the depletion phenotype. The cells are somewhat more elongated but division occurs normally in the sense that one division site is created per cell and that they divide subsequently. See also our answers to reviewer#3 below regarding the different SepF-Scarlet constructs.

9. Does SepF modify FtsZ GTP hydrolysis rates in vitro? The dynamic bundling model proposed here is reminiscent of an interaction between FtsZ and the assembly regulator GpsB in *Staphylococcus aureus*, which bundles but then accelerates disassembly via stimulation of GTP hydrolysis, also serving a checkpoint-like role in vivo (Eswara, et al., 2018). GpsB, however, functions in contrast to most bundling proteins which are largely thought to slow down dynamics or have no effect FtsZ-GTP turnover; therefore it is important to understand how SepF affect FtsZ hydrolysis and dynamics.

We have compared the GTP hydrolysis of FtsZ in the presence and absence of SepF and we have not seen any statistically significant differences between the GTP hydrolysis rate of FtsZ with and without SepF. The results have been added to Supplementary Fig. 7 (panel d) and in the main text (Page 8, lines 1-2) as well as in Supplementary Materials and Methods. Our results are consistent with previous reports showing that SepF from *B. subtilis* does not affect the GTP hydrolysis of FtsZ (Król et Scheffers, J Vis Exp, 2013; Gündoğdu et al, EMBO J, 2011).

Reviewer #3 (Remarks to the Author):

Summary:

The manuscript presented here is a study of the physiological function of *Corynebacterium glutamicum*'s SepF protein. SepF is conserved in many Gram positive bacteria and has been previously shown to interact directly with FtsZCTD, and to have a role in cell division. **This work is supported by many biochemical and in vivo approaches. In this organism that lack homologs of other well studied FtsZ binding proteins and regulators of the Z ring formation understanding the role of SepF in these processes is critical.**

Many of the experiments (especially all the biochemistry) presented in that study are performed well, with the adequate control and presented with enough details that the reader can be confident in the results. These results then support many of the authors conclusions.

The major issues I have with the manuscript concern the experiments in which fusions to fluorescent proteins are used to localize different variants of SepF and FtsZ in vivo. As I detailed below I do think the authors need to improve the experiments presented here and add a few controls to make the reader confident that these experiments supports the authors conclusions. Because of the lack of confidence on the results obtained in these in vivo experiments, I do not believe the model presented at the end of the manuscript is accurate or well supported.

We would like to thank reviewer #3 for the thorough review and constructive criticism of our cellular work. We have performed the control experiments suggested in order to address the concerns of the reviewer. We have also completely revised the model, its description and the figure, as we have realized that the way it was written was confusing. Please find below the detailed changes that we have made to the manuscript.

•Part about SepF being essential:

1. The authors do show convincingly that depletion of SepF impairs cell division (no formation of new septa and of FtsZ rings) but does not block peptidoglycan (PG) synthesis at the poles, leading to some filamentation and then branching of the depleted cells.

I would like the authors to propose an explanation of why branching might occur along the long axis of the cells? How depletion of SepF will lead (after about 6h according to Supl. Fig 2) to new PG synthesis (shown by HADA labelling) on the side of the bacteria, which then seem to lead to the formation of branches. Why is PG synthesis not restricted at the poles even after 6h since no septa are formed?

Ectopic pole formation has been observed in several cell division mutants in mycobacteria, where interfering with cell division seems to delocalize the cell elongation machinery (Gola et al, Mol Microbiol, 2015; Wu et al, J Bacteriol, 2018, Baranowski et al, Microbiol Spectrum, 2019). Even in unrelated bacteria such as *E. coli* branching has been observed when the PG machinery is misplaced (Nilsen et al, Mol Microbiol, 2005; Wells & Margolin, Mol Microbiol, 2012; Potluri et al, Mol Microbiol, 2012). Although the actual mechanisms remain unclear, in *Streptomyces* hyphal branching is known to occur at sites that contain DivIVA, a scaffolding protein for the elongosome assembly (Hempel et al, J Bacteriol, 2008). In *C. glutamicum* the

DivIVA homolog, Wag31, is known to assemble at the poles and septa in normal cells, which correspond respectively to the old and new poles in the daughter cells (i.e. where elongation occurs). Moreover, Wag31 is the most abundant cell division protein in Corynebacteriales (Schuber et al, Cell Host Microbes, 2015), and it is plausible that Wag31 expression could further increase with cell volume in the SepF depleted strain. Under these conditions, an excess of protein may eventually break away from the poles and, in the absence of new septa, nucleate at random loci along the lateral cell wall, thus giving rise to the formation of branches. On the other hand, there could be an as yet to be identified link between FtsZ mislocalization and Wag31. This however would need to be tested experimentally and would be beyond the scope of this paper. We have now included a speculative sentence on branching (Page 5, lines 2-6).

2. When the authors claims that mNeon-FtsZ represent about 3% of the total levels of FtsZ [Bottom of page 5] are they talking of the conditions were there is leakage from the promoter when the cells are grown in sucrose, am I correct? I think the authors should add the fact that the cells Fig 1e or Supl. Fig 5 are also grown in presence of 4% sucrose (I know it is said into the material and methods but it would simplify reading the manuscript if it was clear in the figure legends too).

We have added this information in the legends of Fig. 1 and Supplementary Fig. 4.

3. Can the author check that the expression of mNeon-FtsZ is not affected by the presence of 1% myo-inositol, adding a lane in their western Supplemental Fig4b? My concern is with the pictures presented Fig 1e in which the background level of green fluorescence seems very high and the cells have a lot of cytoplasmic fluorescence not localized into a ring, even a t=0. Is this due to a high expression of the mNeon-FtsZ or a protease processing of the fusion protein leading to a lot of mNeon not fused to FtsZ in the cytoplasm? It is important to know if the level of FtsZ is somehow increased as it may also participate in the inhibition of division phenotype.

We have added the new control in Supplementary Fig. 4 (panel b) to show the expression levels of mNeon-FtsZ in the presence of 0 and 1% myo-inositol are unchanged. The high background fluorescence is mainly due to the auto-fluorescence of *C. glutamicum* when imaged in the green channel, although some degradation of the fusion protein may contribute. Although it is known, the phenomenon of autofluorescence in *C. glutamicum* is poorly reported in the literature (ie Valbuena et al, Microbiology, 2006, data not shown). We have thus included a new Supplementary Fig. 24 of the wild-type strain imaged in the green and red channels to show this phenomenon. We refer to this figure in Materials and Methods (Page 24, line 4-5).

4. Finally, the mNeon-FtsZ, at t=3 in Fig 1e appears to form some kinds of polymers in the cytoplasm of the cells depleted for SepF, but with the green fluo background it is kind of hard to see. Can the authors comments on this? This would suggest that FtsZ at least in vivo could form some kinds of structures (may be bundles) in absence or in condition of depletion of the membrane anchor? This can be important for the authors discussion when they present their model on how SepF may regulate FtsZ bundling.

FtsZ has been observed in small patches outside the Z-ring and it is thought that these are short FtsZ filament structures (Rowlett & Margolin, Biophys J, 2014). We have now included this in the main text (Page 6, lines 9-10).

•Part about Molecular details of SepF interactions:

5. SepF binds to the membrane through its N terminal amphiphatic helix linked to the core of the protein through a long disordered linker. The authors do show that the N terminal first 14 amino acids peptide of SepF sequence that is predicted to form an amphiphatic helix can interact with lipids in vitro (Supl Fig 6) But, a convincing appropriate experiment to show that this sequence is a real membrane targeting sequence in vivo would have been to fuse at the C-terminal of that helix followed by the disordered linker of SepF, a fluorescent protein (GFP, MNeonGFP or Scarlet) and show that this helix can target the fluorescent proteins to the membrane of *C. glutamicum* or even in *E. coli*. Such an experiment would strengthen the authors conclusions.

Following the reviewer's suggestion, we expressed a construct with the first 63 amino acids of SepF (amphiphatic helix + disordered linker) fused to Scarlet. We found that this construct remains cytoplasmic when expressed in *C. glutamicum* (New Supplementary Fig. 12 and Page 9, lines 21-24). Indeed, this construct also remained cytoplasmic when expressed in *E. coli* (see figure below, not included in the manuscript). These results indicate that the amphiphatic helix is unable to direct a protein monomer to the membrane. This is similar to what has been described for *E. coli* FtsA (Loose & Mitchison, Nat Cell Biol, 2013). Therefore, while individual SepF dimers do not bind to the membrane, SepF can nevertheless serve as a membrane tethering factor for a bundle of FtsZ filaments decorated with multiple SepF dimers, due to a higher avidity. Once more, these new results further emphasize why the FtsZ-SepF complex (rather than the individual proteins) are necessary for membrane attachment. This is also in agreement with the SepF mutant unable to bind FtsZ that remains cytoplasmic (Fig. 3c)

6. The authors do show that in vitro, even the SepF Δ ML unable to bind to the membrane can

still interact with FtsZ to bundle FtsZ filaments but that when they mutate 2 residues in SepF that are important in the FtsZCTD binding according to their structure work, SepF lose that ability.

Following these experiments, the authors looked into the *in vivo* properties of these proteins and fused them to the Scarlet fluorescent protein. One comment about this is that I assume the Scarlet protein is fused on the C-terminal side of SepF as this is not described properly in the manuscript. It would help the reader if the authors would at least add in their diagrams of the different SepF species in Fig 3c the position of the linker and Scarlet protein they added to SepF.

We have now stated more clearly where the tags are localized in Materials and Methods (Page 18, lines 9-11) as well as in the main text and figure legends, where the N-ter tag precedes the protein name (mNeon-FtsZ) and the C-ter tag follows the protein name (SepF-Scarlet).

7. I do have many concerns about these SepF-Scarlet fusions, that make me think the fusion protein is not behaving as expected. First the authors say themselves that SepF-Scarlet fusion only partially complemented the SepF depletion strain. Because the cells expressing it are elongated compared to the control cells (Fig 3 and Suppl. Fig 11). When I observe the level of expression of the fusion (Suppl. Fig 11), I am worried that the observed increase in length of the cells is due of the very high level of expression of the fusion instead of its inability to complement the depletion of WT SepF. The authors should definitively show that similar expression of WT SepF (not fused to any tags) are not toxic for the cells and do not cause an increase in size.

To address the reviewer's concerns, we have now added several new controls to distinguish between the effect of the tag and overexpression of SepF without the tag. For the effects of overexpression of SepF without the tag we have created two new strains for ectopic SepF expression from two different promoters: the weak and tightly controlled PgntK promoter and the strong Ptac promoter. The comparative analysis shows that in the case of PgntK we are always situated in low expression levels, with tight repression and almost no leakage to moderate expression upon maximal induction (Supplementary Fig. 20a). This was the reason why we had chosen this promoter in the first place, as it allows to add very little amounts of heterologous proteins. The situation is very different for the Ptac strain, which upon induction gives a ~30 fold increase in protein expression, and the leakage in this system corresponds to the maximal expression in the Pgntk system (Supplementary Fig. 20a). (See also reply to reviewer #2, point 3 above). For the phenotypes we can consider that in the Pgntk strain the cells correspond to WT phenotypes and so does the Ptac strain under non-induced (leakage) conditions. In contrast, the high overexpression from the Ptac system leads to very elongated cells containing just 0 or 1 septum. This information has been included in Supplementary Fig. 20 and in the main text (Page 13, lines 18-22).

Analogously, in order to study the sole effect of the fluorescent tag we have created a new strain where we added the linker and tag to the genomic copy of SepF under the control of its endogenous promoter. In this strain, the SepF-Scarlet fusion localizes at the division plane and the cells are slightly elongated but grow normally and exhibit 0 or 1 septa. We have

added this information in the main text (Page 13, lines 18-22) and in New Supplementary Fig. 20.

8. Also the Western Blots presented Sup Fig 11, show that in addition of the very high levels of expression of these SepF-Scarlet fusion it seems like there is also a lot of proteolysis of the fusion (multiple bands on the lanes when detected with the SepF antibody). The authors should at least do a similar western using an antibody against Scarlet to see what is the actual fluorescent species we are looking at in these cells.

As suggested by the reviewer, we have now done a Western blot using an anti-Scarlet as well as an anti-SepF antibody to check for proteolysis (see new Supplementary Fig. 11). In addition to the full-length fusion protein, we observed a weak band migrating at around 20 kDa for all constructs tested with the anti-Scarlet antibody. From the size of the fragment, we conclude that proteolysis occurs within the mScarlet sequence near its N-terminal region. Therefore, the fluorescence signal in the images mainly comes from the full-length protein, with an additional small contribution of the degradation fragment, assuming it is still functional.

9. I am very concerned because in the cells at T=0 where the authors show sup Fig 11 there is no detectable induction of the fusions proteins by western, there is already enough fluorescence to see localization at the septa of all the proteins fusions (Fig 3c). Then at 6 hours, the fluorescence observed in the cells does not seem to be increased by a lot compared to the t=0 but on the western the expression of the fusion proteins is a lot higher, with multiple bands of different molecular weights (sup Fig 11). In the case of the SepF Δ ML, despite having the same fluorescence levels in the cells that for the other fusion, the expression level of the proteins measured by Western (Sup Fig 11) is not very much. Is that protein unstable, which could explain why it does not complement?

As discussed above the Pgntk promoter permits to use very dilute labels in the cell, however the downside of this is that we have to use very high exposure times to see protein localization under non-induced conditions. This leads also to high fluorescent background. For these reasons, the fluorescent microscopy images were taken with different exposure times to optimize the signal/noise ratio for each condition, and in particular, at t=0 cells were exposed longer since the inducer had no time to increase the level of expression of the fusion protein. Due to this, it is not possible to estimate the expression level of the fusion proteins based on the intensity of the fluorescent signal alone but in concert with the western blot (WB).

Regarding the absence of signal for the SepF Δ ML construct in the WB, this protein is not well recognized by the anti-SepF antibody since most of the immunogenic epitopes seem to be located within the amphipathic helix and the disordered linker (the ML fragment is very well recognized by the anti-SepF antibody, see for instance the band corresponding to the ML-Scarlet construct in the WB revealed with this antibody, Supplementary Fig. 12). For the revised version, we have now repeated the WB using an anti-Scarlet antibody, detecting all the recombinant proteins in its full-length form and we have seen similar levels of expression between all the recombinant proteins. We have included a new Supplementary Fig. 11 containing these results and labelled the relevant bands in each case.

10. For the SepF mutants that do not bind FtsZCTD there is so much cytoplasmic fluorescence that I do not think the authors could see if there is any membrane localization. May be a proteolytic version of the fusion (the band at 25KDa) for example is causing so much cytoplasmic fluorescence that it would make impossible to determine if the fusion that is still full length would bind to the membrane.

The image in Figure 3 for SepF_{K125E,F131A}-Scarlet does indeed show strong cytoplasmic fluorescence. We now demonstrate using an anti-Scarlet antibody (new Supplementary Fig. 11) that this fluorescence is primarily due to the full-length fusion protein and not to a proteolytic version of the fusion, confirming that formation of the FtsZ-SepF complex is necessary for membrane attachment. Moreover, this is in full agreement with our results on the new ML-Scarlet chimera (see our answer to point 5 above), indicating that the amphipathic helix is unable to direct a protein monomer to the membrane.

11. The results with the SepF-Scarlet fusion of the mutant unable to bind to FtsZ are inconsistent with the *in vitro* results the authors talk about later in the manuscript (Fig 5). *In vitro* the authors claim that SepF not bound to FtsZ can attach to the lipid membranes (it makes the SUV tubulated) but then *in vivo* it does not seem to be the case as their mutant is cytoplasmic. I do think the authors should address these differences. May be the fusion on the C-terminal of SepF blocks its ability to polymerize and this could reduce its ability to bind to the membranes (in the literature *E. coli*'s MinD seems to need to be a polymer to be able to bind to the membrane as it was shown that the amphipathic helix of *E. coli*'s MinD is too short to direct a monomer to the membrane). If this is the case for SepF then this could change the interpretation of the results to propose a role of SepF in allowing Z ring formation.

We thank the reviewer for these comments, as they make clear that our original description of the results and the mechanistic model was unclear and confusing. In the revised version, we have now completely rewritten these sections. Briefly, concerning the *in vitro* results reported in Fig. 5, SepF is a dimeric protein prone to polymerize forming rings at high concentration, as seen for the closely related *M. tuberculosis* (Suppl. Fig. 15) and *B. subtilis* (Duman et al, PNAS, 2013) homologs. Membrane binding would then displace the equilibrium towards SepF polymerization (Fig. 5a) in a concentration-dependent manner (Suppl. Fig. 16), while FtsZ binding would do it in the opposite sense, towards depolymerization (Fig. 5b). It follows that the apparent differences noted by the reviewer (i.e. SepF not bound to FtsZ can bind membranes *in vitro*, but not *in vivo*) are indeed a protein concentration issue. At high SepF concentrations (like those used *in vitro*, see Suppl. Fig. 16), SepF binds to (and polymerizes on) the membrane, as shown in Fig. 5a. However, at the much lower physiological concentrations (typically in the nanoMol range for SepF, according to Schuber et al, Cell Host & Microbes, 2015, and our own quantification, see also answer to reviewer 2, point 3, and the new Supplementary Fig. 21) SepF membrane-binding would be undetectable in the absence of FtsZ. This could also account for the cytoplasmic localization of the SepF mutant unable to bind FtsZ (Fig. 3c, left panel). See also our answer to point 5 above.

Regarding the tag, as described in our answer to point 7 above, having the scarlet tag on the sole endogenous copy does not prevent SepF from bringing FtsZ to the membrane and forming a functional Z-ring.

12. Finally the fact that all these fusion proteins are able to be detected at the septum at T=0 in the pictures Fig 3c, strongly suggests that these proteins are still able to be associated with the wildtype SepF (before it is depleted) and despite not being able to localize to the membrane or to interact with FtsZ, these proteins still do not seem to cause major problems at the division site because all the cell size of a t=0 look similar with all the 3 constructs tested.

I do think these experiments with the fusion to scarlet needs to be re addressed before publication as I do not think these confidently support the conclusions the authors are claiming. I would suggest to not even induce the expression of the scarlet fusions as the leak from the gntK promoter is enough to observe the fluorescence (as demonstrated a t=0 in Fig 3c). An alternative would be to perform immunofluorescence experiments using the anti SepF antibody (after affinity purification) instead of using the fusion to Scarlet.

As indicated in our answers above, little amounts of protein are added at time=0, or under non-induced conditions. It can therefore be expected that these proteins, even if not functional, could be incorporated into SepF homodimers or polymers. Indeed we show that this is the case using mass spectrometry for the SepF mutant (main text page 13, lines 9-13). Subsequently, the major aim of inducing overexpression is to look for pathological phenotypes that could give us potential functional hints. In the case of the comparison between WT-SepF-Scarlet and mutant-SepF-Scarlet this can only be done when endogenous SepF is present (as the mutant protein does not localize in the depletion strain). Moreover, we have shown above (see answer to point 7) that, even though the tag is responsible for some cell elongation, it does not affect Z-ring assembly and the cells divide normally. This allows us to compare the behavior between the two constructs with regard to Z-ring assembly and membrane attachment. The concern about the multiple rings coming from SepF overexpression have also been addressed in the answer to point 7 above. When using the Ptac promoter for very high overexpression, the cells do not contain multiple rings, indicating that the multiple rings in Figure 5c do indeed come from the fact that too much mutated SepF has been incorporated in these rings, affecting the FtsZ dynamics (maybe by interrupting patches of treadmilling filaments?) in vivo.

- Part about the role of the C terminal helix of SepF.

13. The authors argue that the structures of SepF unliganded vs SepF bound to the FtsZCTD are the same overall (bottom of page 9). Based on previous structure analysis of other SepF homologs the authors suggest that this C terminal helix $\alpha 3$ might be capping another domain of SepF involved in the polymerization of SepF homodimers in solution. But, the only situations where the authors can observe this type of interaction is in the crystal structure of a SepF deleted for that $\alpha 3$ helix or when it is associated with lipid vesicles. The authors also show that the binding of SepF to the FtsZ CTD reverse that polymerization ability onto lipid membranes. As pointed out above, there are contradictory results between the experiments

done in vivo using the fusion with Scarlet (Figure 3) and these in vitro experiments. In vivo SepF does not seem to be bound to the membrane when it is not interacting with the CTD of FtsZ while in vitro it seems to be able to bind and polymerize at the lipid membrane when it unbound to FtsZ. The authors should discuss this.

We have now entirely rewritten the discussion of the model to clarify the apparent contradictions from our previous version (see Pages 14-15 of the revised manuscript and our answer to point 17 below).

14. Finally the authors try to argue that forming heterodimer of WT SepF with the SepF mutated for its interaction with FtsZ and fused to Scarlet reduce the interaction of a SepF polymer with FtsZ which lead to formation of increased SepF fluorescent rings into the cells when compared to the same experiment using the WT SepF fused to Scarlet as a control. Since as pointed out above the behavior of these scarlet protein does not seem to be well characterized, I think the authors should at least do the following additional control: according to their reasoning if reducing the interaction of SepF with FtsZ lead to a better polymerization of SepF at the membrane to form numerous rings in the cells then expression of wild type SepF-Scarlet into conditions of FtsZ depletion should have the same effect and also leads to many SepF rings in the cells. Such a result would support their conclusions.

We hope that our revised presentation of the data now convinces the reviewer that SepF cannot go to the membrane alone, most likely due to concentration effects. Furthermore, it has been previously shown in *M. smegmatis* that SepF does no longer go to the membrane in the FtsZ-depleted strain (Gola et al, Mol Microbiol, 2015), and therefore it cannot produce a multiring phenotype. Instead, in our model we argue that the multiple rings might come from the fact that mutant SepF perturbs the dynamics of the FtsZ-SepF-membrane system (represented schematically in new Fig. 6c), which is crucial for attaining a functional Z-ring.

15. Another note here to about these Scarlet fusion proteins: According to Sup Fig 18 it looks like the mutant SepF unable to bind to FtsZ and fused to Scarlet is overexpressed a lot more than the WT SepF fused to Scarlet. In addition, that later protein is also processed proteolytically and give a strong additional band at above 37KDa (which is absent in the mutant lane). Such differences of overexpression could explain the fact that the WT SepF may makes less rings in the cells. Can the author comment on this?

As shown above, strong overexpression of WT SepF does not lead to multiple rings. Also as shown above, neither protein overexpression nor inclusion of fluorescent tag account for the observed multiple rings.

•Model:

16. It is very hard to believe that FtsZ filament bundling (which increase the concentration of FtsZ molecules near the SepF molecules that also cause these filaments to be anchored at the membrane) can reduce access to FtsZCTDs. I would like the authors to propose a mechanism for this argument.

As we noted in our answer to point 5 above, our initial description of the model was somewhat confusing in the original manuscript and we have now completely rewritten it. Concerning in particular the specific point above, the crucial issue of our model is not so much that filament bundling can reduce access to FtsZ-CTDs, although this may still be the case because thick bundles can possibly sterically interfere with CTD binding. The crucial feature is rather the highly dynamic nature of the interactions, which can rapidly shift local equilibria between SepF molecules acting as FtsZ-binders or membrane-polymers (or both?). See also our answer to the next point below, in which we describe the main features of the model in more detail.

17. The authors propose in their model (page 14) that the FtsZ bundling and stabilization of FtsZ polymers would increase the local concentration of SepF at the membrane and start SepF polymerization. But they also say that SepF needs to bind FtsZ to go to the membrane (and reciprocally FtsZ binds to SepF to be attached at the membrane). They also say that SepF bound to FtsZ antagonize SepF polymerization, and they show that when an SepF molecule that cannot bind FtsZ is expressed causing a reduced ability of the WT SepF in the cells to binds FtsZ (probably by forming SepF heterodimers), leads to the formation of more SepF polymers (rings). But how does a reduce ability to bind FtsZ also promote recruitment of FtsZ filaments into bundles which is according to the authors model the cause of the start of SepF polymerization?

There are too many contradictory and confusing issues in the way the model is presented, and the authors should do a better job clarifying their model and indicate how it explains their experimental observations.

As noted above, we agree with the reviewer that our original description of the model was confusing. Therefore, we have now completely rewritten this section and illustrated it with a new figure (Fig. 6) in the revised version, we have completely rewritten the discussion (pages 14-16) and we hope that it is now clearer how our proposed model can explain the experimental observations.

In brief, the actual number of SepF copies in the cell is relatively low compared to other cell division proteins such as Wag31 and FtsZ. There are no reliable quantitative estimates available for *C. glutamicum*, but for *M. tuberculosis* there were reported less than 200 SepF molecules per cell (Schubert et al, Cell Host Microbes, 2015), which would correspond to a cellular SepF concentration in the nanoMol range, in agreement with our estimations from Western Blot quantification (see Supplementary Fig. 21). At this concentration, the membrane would be unable to induce SepF polymerization because individual SepF molecules are unable to bind the membrane (see our answers to point 5 and 11 above). For this to happen, a higher local SepF concentration (promoted by FtsZ filament binding) would be required. From our in vitro and in vivo results, the mode of action of SepF could be qualitatively described as schematically showed in new Fig. 6. In an early stage (Fig. 6a), formation of the FtsZ-SepF complex in the cytoplasm leads to FtsZ filament bundling (because of the 2:2 stoichiometry), thus increasing the local SepF concentration around the filaments while precluding its polymerization. At the membrane, oligomeric and non-oligomeric (i.e. dimeric) forms of SepF would coexist, as suggested by the cryo-EM images of the ternary complex (Figs. 5c-d). For instance, some SepF molecules would remain in a dimeric form attached to the bundles, providing enough amphipathic peptides to sustain membrane

tethering (Fig. 6b, top), while others would detach from the bundles to form membrane-bound polymers that will play an active role in remodeling the lipid bilayer for septum formation (Fig. 6b, bottom). Such a redistribution is made possible by the highly dynamic nature of the FtsZ-SepF-membrane interactions and the opposite effects of competitive filament/membrane binding on SepF oligomerization. How the SepF-FtsZ complexes are specifically localized to the midcell and whether there are additional regulatory factors that are involved in the oligomerization state of SepF remains to be elucidated. These could be unidentified divisome components or else, post-translational modifications such as protein phosphorylation, known to play an important role in the regulation of actinobacterial cell division and morphogenesis (Manuse et al, FEMS Microbiol Rev, 2015).

Of course, many aspects of this mechanistic model remain unknown, such as the directionality of FtsZ filaments in SepF-induced bundles, the actual relationship between the linear FtsZ filaments and membrane-attached SepF polymers (which may be perpendicular to each other, as suggested by Duman et al, PNAS, 2013, for *B. subtilis* SepF), or yet the identification of additional factors that might be required to stabilize (or destabilize) the FtsZ-SepF-membrane system in later stages of divisome assembly and function.

Concerning the reviewer's question about the mutant form of SepF unable to bind FtsZ, our model correctly predict that this protein will be cytoplasmic when expressed in the context of the SepF depleted strain (Fig. 3c, left panel), because it is unable to bind FtsZ filaments. Instead, when expressed in the context of the WT strain, wild-type/mutant SepF heterodimers can still bind to FtsZ filaments (although they will not contribute to the bundling effect because of the 1:2 SepF:FtsZ stoichiometry) and would therefore be brought to the membrane. Once there, SepF mutants would behave as the wild-type protein for membrane-supported polymerization. In other words, according to our model the introduction of mutated SepF in a WT SepF background would shift the dynamic equilibrium of FtsZ-binding SepF dimers vs ring-formers SepF polymers to favour the latter, which could account for the observation of multiple rings.

Reviewers' comments:

Reviewer #2 (Remarks to the Author):

The authors satisfactorily addressed my concerns in the revised manuscript.

Reviewer #3 (Remarks to the Author):

The manuscript presented here is the resubmission of a previous version. In this manuscript the authors study the physiological role of *Corynebacterium glutamicum* SepF which appears to be the only conserved cell division protein interacting with FtsZ in this organism. In this manuscript, the authors showed that SepF is essential, interacts with FtsZ and behave as a membrane anchors for FtsZ filaments at the division site. Based on their results they then propose a mechanistic model for how SepF-FtsZ interaction may regulate Z ring assembly in this Actinobacteria.

Overall the manuscript is clearly written but at many places, the description of the experiments in the figure legends are not enough to fully understand how these were performed (strain background or which inducer was used are missing) and may need some work before publication [for an example, see my comment below about Fig Sup12 in which we have to assume the experiment is done in a strain WT for SepF].

In this resubmission, the authors have addressed many of the reviewer comments and have provided new controls and experiments in order to strengthen their work.

I am convinced by most of the experiments presented here that SepF is essential, that it interacts with FtsZ through a pocket formed by the homodimer and that it is probably the membrane anchor for FtsZ.

Based on the experiment presented here, I think but I am not convinced that SepF may need to be a multimer of homodimers to bind to the membrane and that it may reach that state in the cells only because it may be "locally enriched" by its interactions with FtsZ polymers at the division site [because these Z polymers are only "allowed" (by an unknown positioning mechanism to form there)]. Thus, this codependence of FtsZ polymers and SepF interaction in promoting SepF ability to bind to the membrane may be part of the mechanism by which SepF serves as an anchor for FtsZ rings at the division sites.

Still, I am very doubtful of the part of the model where the homodimers of SepF forming a polymer required for its binding to the membrane is counteracted by the binding of SepF to FtsZ. I can't imagine this process (even if it is very dynamic) being an efficient way to anchor FtsZ polymers to the membrane in the cells where there are 20 times more FtsZ than SepF molecules.

The authors responses to my concerns about the Scarlet fusions are not satisfying me enough to give me confidence in their in vivo results. The fusions (beside the one with the deletion of ML) are a lot more expressed from their Pgntk than the level of endogenous SepF in the cells (like 20 to 25 times when you look at the western shown Sup Fig 19 and this is even without including the degradation products). Despite this huge overexpression of WT SepF-Scarlet the cells are only slightly elongated. In Figure Sup 20 panel b the authors indicate that an over expression of SepF by 28 times cause phenotype where the FtsZ rings are gone from the cells. Do that mean that the scarlet fusion to SepF is not fully functional? May be the fusion of Scarlet to the Cterminus of SepF prevent the function of that Helix 3 important for regulation of the SepF homodimer oligomerization (which could explain the difference of results between their in vitro results with untagged SepF and the in vivo results with the scarlet fusions). Also, when SepF-Scarlet fusion complements the depletion of SepF, this complementation could be due to the degradation product that is not fluorescent but provide enough of functional SepF to allow cell growth (there is so much degradation and the degradation product at higher than the level of endogenous SepF)? I am still not convinced that the experiments done with their Scarlet fusions are of a quality good enough to

be giving trustable results.

Sup Fig12: Because of their lack of details in the legend it is not clear if they express the SepFML into a strain wild type for SepF but why can they detect the chromosomal SepF in the lane #2 but not in the lane #1 of their Western blots using the antibody against SepF? I assume it should be there too. In addition there is so much degradation product of the fusion recognize by the scarlet antibody at around 30KDa. Is that degradation product causing so much background fluorescence that a membrane halo of fluorescence by the fusion of the ML region of SepF to scarlet could be missed if there was one?

Page 10 line #1 and 2, the authors claim the binding to the membrane of the ML sequence is not detected for FtsA MTS alone, but this is inaccurate as a GFP fused to FtsA's MTS is targeted to the membrane as illustrated Fig3 panel B of Pichoff and Lutkenhaus 2005 Mol. Microbiol. In that same publication Pichoff and Lutkenhaus showed that FtsA does not need to bind to FtsZ to interact with the membrane. FtsA requires FtsZ (which localization in the cells is restricted by the effect of Min and SlmA proteins) to localize at the septum, because FtsA interacts with FtsZ and it is where FtsZ is localized, not because the interaction with FtsZ promotes FtsA to go to the membrane, there is a difference.

Instead of using their fusion to Scarlet, I do think the authors should look at the effects of expressing untagged SepF constructs and mutants in SepF depleted cells (look for complementation) and in WT cells where they could follow their effect on the WT SepF ring [by using the strain where the SepF-Scarlet fusion is on the chromosome]. Are the SepF truncated for ML domain or Helix 3 or the double mutant K125/F131 dominant negative? What kind of effects they have on localization of WT SepF or of FtsZ do they have. These information could then be discussed to see if they fit the author's model and would provide more comparable results between the constructs. Also since the Ptac promoter seems to work pretty well in their organism they could use different level of IPTG to vary level of induction of their proteins and compare if a mutant impaired for interaction with FtsZ need to be overexpressed more than a mutant that does not go to the membrane in order to be localize at the Z ring for exemple (Since a mutant that do not interact with FtsZ is less likely to be "concentrated in the cells" enough to be allowed to polymerize with the WT SepF proteins [aka they should not be localized at the same place in the cells, wt concentrated near FtsZ bundles while the SepF mutant for FtsZ interaction should be diluted in the entire cytoplasmic space] if you believe the author's model).

It is not clear to me either why would a mutant that do not interact with FtsZ, and according to the author's model could not be concentrated enough at one location in the cell to form oligomers of homodimers (even if it is a dimer of a WT and mutant version of the SepF protein which should have a reduced interaction with FtsZ) to be able to polymerize into more rings (as seen Fig 5) than the WT SepF (which according to the same model depend on the ability of its interaction with FtsZ bundle to be concentrated enough in one place to start to be recruited at the membrane)?

This brings me to the point that the model (even rewritten in that version) still does not make sense to me. How can a protein that anchors the FtsZ polymers to the membrane, need the interaction with the FtsZ to be able to attached at the membrane (because this interaction will increase its local concentration and induces its ability to oligomerize), while at the same time, that same interaction with FtsZ reduce SepF ability to oligomerize, reducing its ability to be incorporated into a polymer which is required for SepF to be attached to the membrane (hence interfering with its ability to anchor the FtsZ polymers bundle at the membrane)? And also how can this be done in cells where FtsZ is 20 time more abundant than the SepF protein?

Reviewer #3 (Remarks to the Author):

The manuscript presented here is the resubmission of a previous version. In this manuscript the authors study the physiological role of *Corynebacterium glutamicum* SepF which appears to be the only conserved cell division protein interacting with FtsZ in this organism. In this manuscript, the authors showed that SepF is essential, interacts with FtsZ and behave as a membrane anchors for FtsZ filaments at the division site. Based on their results they then propose a mechanistic model for how SepF-FtsZ interaction may regulate Z ring assembly in this Actinobacteria.

Overall the manuscript is clearly written but at many places, the description of the experiments in the figure legends are not enough to fully understand how these were performed (strain background or which inducer was used are missing) and may need some work before publication [for an example, see my comment below about Fig Sup12 in which we have to assume the experiment is done in a strain WT for SepF].

The reviewer states that in many figure legends the strain or inducer conditions are missing. This was the case for supplementary Figures 12 and 20 (out of a total of 31 Figures in the manuscript), both of which have now been completed.

In this resubmission, the authors have addressed many of the reviewer comments and have provided new controls and experiments in order to strengthen their work. I am convinced by most of the experiments presented here that SepF is essential, that it interacts with FtsZ through a pocket formed by the homodimer and that it is probably the membrane anchor for FtsZ.

Based on the experiment presented here, I think but I am not convinced that SepF may need to be a multimer of homodimers to bind to the membrane and that it may reach that state in the cells only because it may be “locally enriched” by its interactions with FtsZ polymers at the division site [because these Z polymers are only “allowed” (by an unknown positioning mechanism to form there)]. Thus, this codependence of FtsZ polymers and SepF interaction in promoting SepF ability to bind to the membrane may be part of the mechanism by which SepF serves as an anchor for FtsZ rings at the division sites.

We would like to correct the phrase of Reviewer 3: “that SepF may need to be a **multimer of homodimers** to bind to the membrane”. This is not what we state in the paper. We say that SepF needs to be enriched on the FtsZ filaments in order to provide enough binding motifs to go to the membrane (i.e. an avidity effect, for a nice description on this effect see the latest review by Du & Lutkenhaus, 2019). At this stage SepF is not in an oligomeric form. This is clearly stated in the main text as well as in Figure 6.

Still, I am very doubtful of the part of the model were the homodimers of SepF forming a polymer required for its binding to the membrane is counteracted by the binding of SepF to FtsZ. I can't imagine this process (even if it is very dynamic) being an efficient way to anchor FtsZ polymers to the membrane in the cells where there are 20 times more FtsZ than SepF molecules.

We don't fully understand the point raised by the reviewer. First, we experimentally showed that the addition of FtsZ depolymerizes SepF (Fig. 5b), but as shown in Fig. 5d the FtsZ filaments still remain bound to the membrane (liposomes). These observations are fully compatible with a dynamic model in which there is a continuous rearrangement (involving both SepF polymerization and depolymerization coupled to membrane binding), in a similar way as suggested for FtsA/FtsZ (see Loose and Mitchinson, 2014). It is important to emphasise that in both cases SepF dimers or polymers are attached to the membrane.

The authors responses to my concerns about the Scarlet fusions are not satisfying me enough to give me confidence in their in vivo results. The fusions (beside the one with the deletion of ML) are a lot more expressed from their Pgntk than the level of endogenous SepF in the cells (like 20 to 25 times when you look at the western shown Sup Fig 19 and this is even without including the degradation products). Despite this huge overexpression of WT SepF-Scarlet the cells are only slightly elongated. In Figure Sup 20 panel b the authors indicate that an over expression of SepF by 28 times cause phenotype where the FtsZ rings are gone from the cells. Do that mean that the scarlet fusion to SepF is not fully functional?

In the first round of revision, we have already shown that SepF-Scarlet construct is fully functional because we replaced the **genomic** copy of *sepF* by the *sepF-scarlet* fusion and the strain is wild-type-like (see supplementary Figure 20d). In a more general note, we are fully aware that fluorescent fusion proteins are artefactual and cannot reflect exact *in vivo* conditions. However, they have been used in cell biology for decades to help address important biological questions. In all cases we compare fusion proteins that are comparable (for example WT vs mutant, where expression levels, fusion partners and degradation levels are the same and the only difference is the mutation introduced). As we are aware of the limitations of the cell biology data and possible artefacts, we coupled them to solid biophysical and structural data.

May be the fusion of Scarlet to the C-terminus of SepF prevent the function of that Helix 3 important for regulation of the SepF homodimer oligomerization (which could explain the difference of results between their in vitro results with untagged SepF and the in vivo results with the scarlet fusions). Also, when SepF-Scarlet fusion complements the depletion of SepF, this complementation could be due to the degradation product that is not fluorescent but provide enough of functional SepF to allow cell growth (there is so much degradation and the degradation product at higher than the level of endogenous SepF)? I am still not convinced that the experiments done with their Scarlet fusions are of a quality good enough to be giving trustable results.

The full-length SepF-Scarlet fusion can clearly complement the *sepF*-depleted mutant, because we were able to replace the genomic copy of *sepF* by the *sepF-scarlet* fusion. In this wild-type-like strain, no degradation of the fusion protein was observed by Western Blot (supplementary Figure 20d).

Sup Fig12: Because of their lack of details in the legend it is not clear if they express the SepF_{ML} into a strain wild type for SepF but why can they detect the chromosomal SepF in the lane #2 but not in the lane #1 of their Western blots using the antibody against SepF? I assume it should be there too. In addition there is so much degradation product of the fusion recognize by the scarlet antibody at around 30KDa. Is that degradation product causing so much background fluorescence that a membrane halo of fluorescence by the fusion of the ML region of SepF to scarlet could be missed if there was one?

We have corrected this. As now stated in the legend to Supplementary Figure 12, the SepF_{ML}-Scarlet construct is expressed in the wild-type strain named WT-*P_{gntK}-sepF_{ML}-scarlet*. Also, a low intensity band is visible for endogenous SepF when revealed with the anti-SepF antibody in lane #1, whereas the higher intensity of lane #2 does indeed correspond to endogenous SepF plus a SepF_{ML}-containing degradation fragment that unfortunately co-migrates with SepF (as seen in the anti-Scarlet WB of the same figure). We have now increased the contrast of the Western Blot to see more clearly the endogenous band in lane #1 and the degradation fragment is now labelled in the figure.

We do not agree that the cytosolic degradation product would hide the membrane localization, as we are not in saturating conditions of fluorescence and the membrane and cytosolic stain can be differentiated even at high levels of background and low full-length protein (as seen for WT-*P_{gntK}-sepF* in sucrose conditions). Also, the membrane at the division plane shows no or very low diffuse signal, which would not be expected if the localization at the membrane were present (you would see the double of the intensity). We are confident that there is no membrane localization with the ML fusion construct.

Page 10 line #1 and 2, the authors claim the binding to the membrane of the ML sequence is not detected for FtsA MTS alone, but this is inaccurate as a GFP fused to FtsA's MTS is targeted to the membrane as illustrated Fig3 panel B of Pichoff and Lutkenhaus 2005 Mol. Microbiol. In that same publication Pichoff and Lutkenhaus showed that FtsA does not need to bind to FtsZ to interact with the membrane. FtsA requires FtsZ (which localization in the cells is restricted by the effect of Min and SlmA proteins) to localize at the septum, because FtsA interacts with FtsZ and it is where FtsZ is localized, not because the interaction with FtsZ promotes FtsA to go to the membrane, there is a difference.

We thank the reviewer for raising this point to our attention. We agree that our sentence was inaccurate, as FtsA can indeed localize to the membrane independently of FtsZ (although its localization at the septum is still dependent on FtsZ, as shown in Pichoff and Lutkenhaus, 2005). We have now modified this sentence accordingly in the text (p.10 Lines 1-2).

Instead of using their fusion to Scarlet, I do think the authors should look at the effects of expressing untagged SepF constructs and mutants in SepF depleted cells (look for complementation) and in WT cells where they could follow their effect on the WT SepF ring [by using the strain where the SepF-Scarlet fusion is on the chromosome]. Are the SepF truncated for ML domain or Helix 3 or the double mutant K125/F131 dominant negative? What kind of effects they have on localization of WT SepF or of FtsZ do they have. These information could

then be discussed to see if they fit the author's model and would provide more comparable results between the constructs. Also since the Ptac promoter seems to work pretty well in their organism they could use different level of IPTG to vary level of induction of their proteins and compare if a mutant impaired for interaction with FtsZ need to be overexpressed more than a mutant that does not go to the membrane in order to be localize at the Z ring for exemple (Since a mutant that do not interact with FtsZ is less likely to be "concentrated in the cells" enough to be allowed to polymerize with the WT SepF proteins [aka they should not be localized at the same place in the cells, wt concentrated near FtsZ bundles while the SepF mutant for FtsZ interaction should be diluted in the entire cytoplasmic space] if you believe the author's model).

We are not completely sure we follow the argument of the reviewer here. We have chosen to use Scarlet fusions because a primary goal was to carry out localization studies, and we did the necessary controls to justify them (see our answers above). We have positively addressed all experimental controls required in the first round of revision, and we are fully aware that we could keep introducing additional controls *ad infinitum*. However, we think that this would likely add little new information in an already dense manuscript. Moreover, a categorical validation of our proposed model is clearly beyond the scope of the present manuscript.

It is not clear to me either why would a mutant that do not interact with FtsZ, and according to the author's model could not be concentrated enough at one location in the cell to form oligomers of homodimers (even if it is a dimer of a WT and mutant version of the SepF protein which should have a reduced interaction with FtsZ) to be able to polymerize into more rings (as seen Fig 5) than the WT SepF (which according to the same model depend on the ability of its interaction with FtsZ bundle to be concentrated enough in one place to start to be recruited at the membrane)? This brings me to the point that the model (even rewritten in that version) still does not make sense to me. How can a protein that anchors the FtsZ polymers to the membrane, need the interaction with the FtsZ to be able to attached at the membrane (because this interaction will increase its local concentration and induces its ability to oligomerize), while at the same time, that same interaction with FtsZ reduce SepF ability to oligomerize, reducing its ability to be incorporated into a polymer which is required for SepF to be attached to the membrane (hence interfering with its ability to anchor the FtsZ polymers bundle at the membrane)? And also how can this be done in cells where FtsZ is 20 time more abundant than the SepF protein?

In these paragraphs the reviewer raises questions about our proposed mechanistic implications. However, the working model we propose in the discussion for membrane-SepF-FtsZ interactions, albeit speculative and necessarily incomplete (as only a minimal set of positive and negative regulators of cytokinesis are as yet known in *Actinobacteria*) is fully compatible with all experiments reported in the manuscript. We agree with the reviewer that future experiments will be required to understand the precise molecular details of the ternary complex assembly and disassembly.

REVIEWERS' COMMENTS:

Reviewer #3 (Remarks to the Author):

The manuscript presented here is of significant interest for the field and present some very interesting findings that will bring a new understanding of the FtsZ ring formation in *Cornybacterium*.

In this third round of review not much is changed from the previous round. I am still skeptic about the model in which SepF binding to FtsZ filaments allow SepF to multimerize and bind to the membrane despite the fact that FtsZ binding to SepF cause SepF multimer to depolymerize. I do not clearly understand then how SepF can serve as a membrane anchor to the FtsZ polymers in the cells (noting that the concentration of FtsZ is 20 times higher than the one of SepF).

As I expressed in my previous reports, I am still concerned that the fusion of Scarlet to the C terminal of SepF may affect some of SepF functions and that some of the phenotype observed might be due to processed products of that fusion (degradations fragments) instead of being caused by the full length fusion. As an example, an overexpression of SepF by 28 times causes the cells to be very elongated and the Z rings to disappear but a similar overexpression of SepF-Scarlet fusion only cause a mild elongation of the cells. I am afraid that the fusion on the C terminus of SepF may affect the role of SepF's Helix H3 which may cause some mis interpretation of the results.

Despite my issues with the Scarlet fusions and the model, but because this manuscript also presents other very interesting results that are well supported by the experiments, then I would recommend that the manuscript be published so then the readers can make their own opinion of the part of the manuscript I have issue with.

REVIEWERS' COMMENTS:

Reviewer #3 (Remarks to the Author):

The manuscript presented here is of significant interest for the field and present some very interesting findings that will bring a new understanding of the FtsZ ring formation in *Corynebacterium*.

In this third round of review not much is changed from the previous round. I am still skeptic about the model in which SepF binding to FtsZ filaments allow SepF to multimerize and bind to the membrane despite the fact that FtsZ binding to SepF cause SepF multimer to depolymerize. I do not clearly understand then how SepF can serve as a membrane anchor to the FtsZ polymers in the cells (noting that the concentration of FtsZ is 20 times higher than the one of SepF).

We would like to insist that our postulated model, though it agrees with our experimental data, is necessarily speculative (as stated in the manuscript) and intends to inspire further experimental work for validation or refutation. As a matter of fact, a recent (unreviewed) preprint by Wenzel et al (bioRxiv, doi: <https://doi.org/10.1101/771139>) seems to lend some support to our model. The authors observed a correlation between SepF ring diameters and septa thickness in different bacterial species, leading them to suggest that “SepF polymerizes into molecular clamps on the leading edge of nascent septa”, in much the same way as we propose in our model (Fig. 6b).

As I expressed in my previous reports, I am still concerned that the fusion of Scarlet to the C terminal of SepF may affect some of SepF functions and that some of the phenotype observed might be due to processed products of that fusion (degradations fragments) instead of being caused by the full length fusion. As an example, an overexpression of SepF by 28 times causes the cells to be very elongated and the Z rings to disappear but a similar overexpression of SepF-Scarlet fusion only cause a mild elongation of the cells. I am afraid that the fusion on the C terminus of SepF may affect the role of SepF's Helix H3 which may cause some mis interpretation of the results.

We agree with Reviewer 3 that the C-terminal fusion does affect the function of SepF to a certain extent, and this would indeed be expected if our hypothesis of the C-ter acting as a molecular switch is true. We have now explicitly mentioned this caveat in the manuscript (page 9, lines 7-9), which might explain the phenotypic differences noted by the reviewer between SepF and SepF-Scarlet.

Our main point, however, is that when we compare two strains where the only difference is a mutation in SepF and not the Scarlet fusion, which remains the same (as we do in page 9, first paragraph, and also in page 13, second paragraph), the observed differences are primarily due to the mutation, and thus provide some important information on the function.

Despite my issues with the Scarlet fusions and the model, but because this manuscript also presents other very interesting results that are well supported by the experiments, then I would recommend that the manuscript be published so then the readers can make their own opinion of the part of the manuscript I have issue with.